# DiffuseGuide: Guiding Diffusion Models Made Easy

## Abstract

Despite advancements in conditional generation using diffusion models, conditional generation remains affected by training cost, generalizability, and speed. Training free conditional generation assists in these avenues where a model can be steered to adhere any particular condition through inference time optimization. However, existing techniques often rely on computationally intensive backpropagation through the diffusion network to estimate the guidance direction, compounded by the need for meticulous parameter tuning tailored to individual tasks. Although some recent works have introduced minimal-compute methods for linear inverse problems, a generic, lightweight guidance solution for both linear and non-linear guidance problems is still missing. To this end, we propose *DiffuseGuide*, a method that enables inference-time guidance without compute-heavy backpropagation through the diffusion network. The key idea is to approximate the guidance direction with respect to the current sample, thereby removing the backpropagation operation. Moreover, we propose an empirical guidance scale that works for a wide variety of tasks, thus removing the need for handcrafted parameter tuning. We further introduce an effective, lightweight augmentation strategy that significantly boosts performance during inference-time guidance. We present experiments using DiffuseGuide on multiple linear and non-linear tasks across multiple datasets and models to show the effectiveness of the proposed modules.

## 1 Introduction

Generative modeling with Denoising Diffusion Probabilistic Models (DDPMs) Sohl-Dickstein et al. (2015); Ho et al. (2020); Dhariwal & Nichol (2021); Song et al. (2021b) has improved massively over the past few years. Multiple works have extended diffusion models to text-to-image synthesis Balaji et al. (2022); Rombach et al. (2021); Saharia et al. (2022b), 3D synthesis Poole et al. (2022); Jun & Nichol (2023), video generation Ho et al. (2022); Blattmann et al. (2023); Wu et al. (2023a), as well as conditioning to solve inverse problems. Moreover, like conditional generative adversarial networks (GANs) Goodfellow et al. (2020); Arjovsky et al. (2017), DDPMs can be adapted to tasks based on a label Rombach et al. (2021); Dhariwal & Nichol (2021) or visual prior–based conditioning Saharia et al. (2022a). However, like conditional GANs Wang et al. (2018); Radford et al. (2015), DDPMs also need to be trained with annotated pairs of labels and instructions for satisfactory results. This poses a limitation in many cases where there is a lack of paired data to train large diffusion models. For this reason, there has been recent interest in models that can perform conditional generation without the need for explicit training Yu et al. (2023); Chan et al. (2016); Nguyen et al. (2017); Graikos et al. (2022).

Progressing in this direction is prior research in plug-and-play models. First introduced in Nguyen et al. (2017), the initial research on plug-and-play models Nguyen et al. (2017); Graikos et al. (2022) enabled conditional sampling from GANs trained with unlabeled data. For this, a pretrained classifier Simonyan & Zisserman (2014); Hossain et al. (2019) or a captioning model was used to estimate the deviation between the GAN-generated image and a given label; based on this deviation, the GAN input noise was modulated until the generated sample satisfied the given text or class label. A similar approach has been attempted for diffusion models to facilitate conditional sampling from unconditional diffusion models: classifier guidance Dhariwal & Nichol (2021); Graikos et al. (2022), where a noise-robust classifier is trained along with the diffusion model to guide sampling toward a particular direction. However, classifier guidance brings in the computational costs of

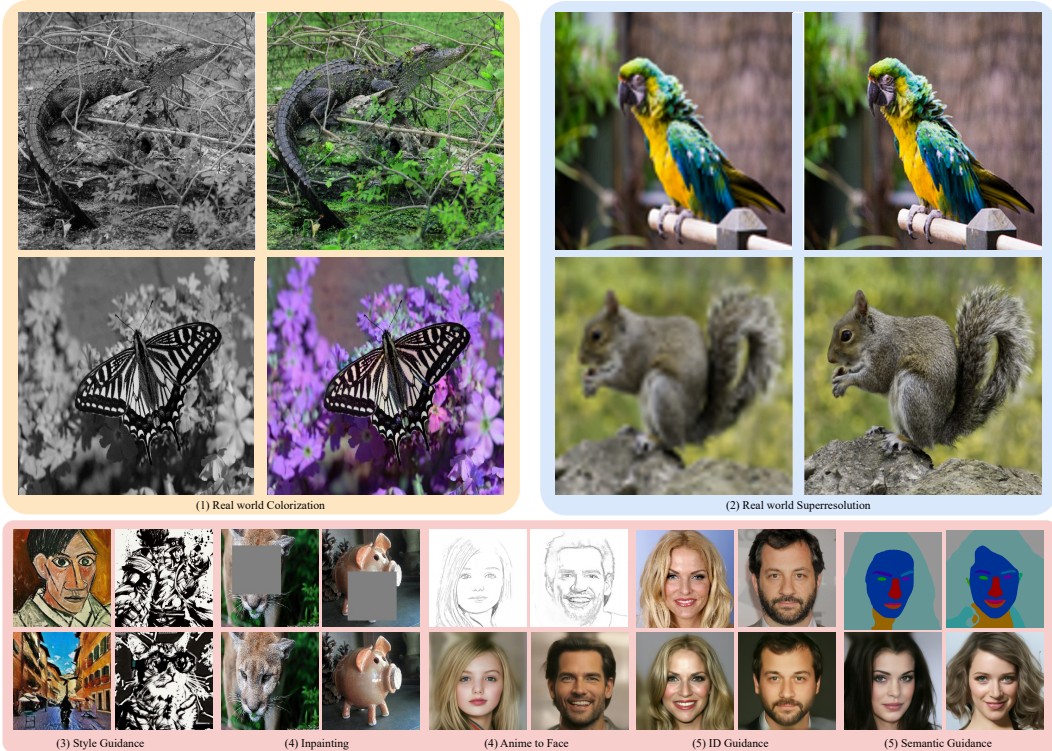

Figure 1: An illustration of the different applications of our method. We utilize a pretrained diffusion model to generate images satisfying a predefined condition without any training, backpropagation through the diffusion U-Net, or any hand-crafted parameter tuning. We present results on (1) real-world colorization, (2) real-world super-resolution, (3) style-guided text-to-image generation, (4) inpainting, (5) sketch-to-face synthesis, (6) face ID guidance, and (7) face semantics–to–face.

training a classifier, which is often undesirable. Some recent works have performed conditional generation without explicit training for the condition by utilizing the implicit guidance capabilities of the diffusion model Chung et al. (2023b); Yu et al. (2023); Nair et al. (2023); Bansal et al. (2023); Chung et al. (2023a). Diffusion Posterior Sampling (DPS) Chung et al. (2023b) proposed using an $L_2$ loss to solve linear inverse problems with unconditional diffusion models, but often requires many sampling steps for photorealistic results. Freedom Yu et al. (2023) proposed using general loss functions during sampling to achieve training-free conditional sampling. Variants of DPS have also been proposed Song et al. (2023). All the aforementioned loss-guided posterior sampling techniques involve a guidance function at each timestep that requires backpropagation through the diffusion U-Net.

Recently, He et al. (2023) proposed Manifold-Preserving Guided Diffusion Models (MGD) that remove the need for backpropagating through the diffusion U-Net by performing gradient descent with respect to the Minimum Mean-Square Error (MMSE). Although MGD works remarkably well for linear tasks that require more guidance toward the later stages of the sampling process, it may fail in tasks where guidance is needed earlier—for example, face semantics–to–image and sketch–to–image—where stronger guidance is required from much earlier stages. Moreover, like Yu et al. (2023); Nair et al. (2023), MGD also requires a handcrafted parameter on a case-by-case basis. Hence, a generic, lightweight method that works well for both linear and non-linear guidance functions is still missing. The need to find a handcrafted guidance parameter on a case-by-case basis remains an open challenge.

In this paper, we introduce a new framework that can adaptively perform zero-shot generation using diffusion models without manual intervention. We found a simple fix to the problem during the initial timesteps of diffusion: utilize the gradient with respect to the diffusion output noise. Combined with guidance with respect to the MMSE estimate, this combination generalizes well to tasks that require early guidance. Figure 2 visualizes our approach relative to existing works. Using the correction term together with the correction with respect to the MMSE estimate significantly boosts performance in non-linear tasks. We present the corresponding results in Section 6.

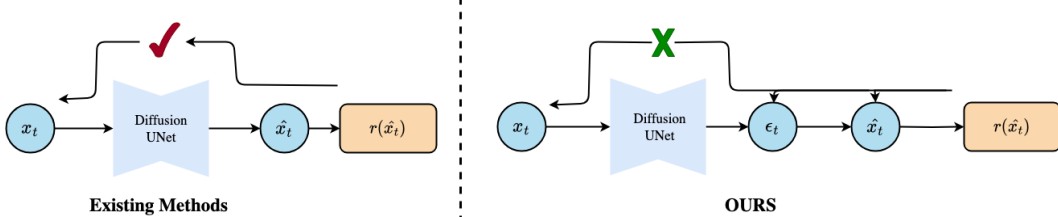

Figure 2: An illustration of the difference between existing methods and ours. Existing works backpropagate through the diffusion network to perform guidance at each timestep, whereas we compute gradients with respect to the MMSE estimate and the predicted noise, thereby bypassing the expensive backpropagation operation.

Moreover, we treat energy-based inference-time guidance Chung et al. (2023b); Yu et al. (2023) as stochastic gradient optimization of the MMSE estimate and the noise present in the image. This formulation enables us to leverage recent research in parameter-free learning Defazio & Mishchenko (2023); Ivgi et al. (2023) to develop a dynamic step-size schedule. This step size adapts to the initial noise seed input of the diffusion model and the guidance functions, thereby removing the need for manual parameter tuning at inference time. Motivated by the effectiveness of differentiable augmentations when training GANs Zhao et al. (2020), we found that utilizing multiple levels of matched differentiable augmentations on the MMSE estimate and the guidance reference significantly improves sampling quality, enabling very high-quality sampling with a low number of guidance steps. We present an overview of the different applications of our method in Figure 1. Specifically, we present results using Stable Diffusion Rombach et al. (2021), unconditional diffusion models released by Nichol & Dhariwal (2021) for $256 \times 256$ guidance, and class-conditional diffusion models for high-resolution $512 \times 512$ conditional synthesis. The different functionalities of DiffuseGuide are tabulated in Figure 2.

We present experiments on publicly released models on generic images, face images, and Stable Diffusion to show the relevance of our method. We focus on the tasks of (1) inpainting, (2) super-resolution, (3) colorization, (4) Gaussian deblurring, (5) semantic label–to–image generation, (6) face sketch–to–image, and (7) ID guidance and identity generation, and we beat existing benchmarks that utilize diffusion models for these tasks, obtaining a significant performance boost over existing loss-guided methods. To summarize, our contributions are:

## 2 RELATED WORK

### 2.1 TRAINING-FREE CONDITIONAL SAMPLING USING DIFFUSION MODELS

Recently, there has been a rise in works that propose utilizing unconditional diffusion models for conditional sampling Bansal et al. (2023); Chung et al. (2023c); Kawar et al. (2022); Nair et al. (2023). Earlier works proposed solving linear inverse problems using diffusion models with priors dependent on the inverse transform of the degradation. Diffusion Posterior Sampling (DPS) Chung et al. (2023b) considered the degradation to be conditioned on a Gaussian distribution at any intermediate timestep and derived an $L_2$ regularization at each intermediate timestep to solve linear inverse problems. Freedom Yu et al. (2023) explored an energy-based perspective and extended guidance to non-linear functions using general loss functions. Universal diffusion guidance Aggarwal et al. (2018) extended this guidance process to Stable Diffusion and improved performance using forward–backward guidance. More recent works, such as manifold-guided diffusion He et al. (2023), further constrained the manifold space by projecting the latent space alone. Steered diffusion Nair et al. (2023) guided implicit predictions for non-linear functions, utilizing a hard constraint for normal functions and proposing a plug-and-play module to improve performance.

Table 1: Capabilities of DiffuseGuide versus existing methods for inference-time guidance.

| Method | Zeroth order | Linear Tasks | Non-Linear Tasks | Automatic scaling |
|---|---|---|---|---|
| DPS Chung et al. (2023a) | ✗ | ✓ | ✗ | ✗ |
| πGDM Song et al. (2022) | ✗ | ✓ | ✗ | ✗ |
| Freedom Yu et al. (2023) | ✗ | ✗ | ✓ | ✗ |
| MGD He et al. (2023) | ✓ | ✓ | ✗ | ✗ |
| OURS | ✓ | ✓ | ✓ | ✓ |

## 3 BACKGROUND

### 3.1 PERTURBED MARKOVIAN KERNEL FOR DIFFUSION TRANSITION

For conditional generation tasks using an unconditional diffusion model, ideally the model would predict intermediates closer to the condition. Let $r(x_t, y)$, where $x_t$ denotes the noisy latent variable at diffusion timestep $t$ and $y$ denotes the conditioning signal (e.g., label or image), give a measure of the distance between an intermediate $x_t$ and the condition $y$ and be a positive, bounded function. Hence, in the reverse process, the diffusion trajectory should proceed through distributions with a higher probability of being closer to the desired cases. We model these trajectory intermediate distributions with

$$\hat{p}(x_t) = p(x_t)r(x_t, y), \tag{1}$$

where $p(x_t)$ is the unconditional marginal distribution of $x_t$ and $\hat{p}(x_t)$ is the perturbed distribution.

Sohl-Dickstein et al. Sohl-Dickstein et al. (2015) first proposed the use of Markovian kernels to estimate the distribution of diffusion intermediates. Specifically, given the state $x_t$ at the equilibrium of the training process for a diffusion model, the distribution at timestep $t-1$ can be estimated as

$$p(x_{t-1}) = \int p(x_t)p_\theta(x_{t-1}|x_t)\,dx_t, \tag{2}$$

where $x_{t-1}$ denotes the latent variable at the next reverse timestep and $p_\theta(x_{t-1}|x_t)$ is the Gaussian transition kernel parameterized by the U-Net $\theta$.

The kernel $p(x_{t-1}|x_t)$ is a Gaussian distribution whose mean can be estimated using the diffusion U-Net and $x_t$. To estimate a perturbed kernel $\hat{p}(x_{t-1}|x_t)$, the perturbed distribution can be modeled as

$$p(x_{t-1})r(x_{t-1}, y) = \int r(x_t, y)p(x_t)\hat{p}_\theta(x_{t-1}|x_t)\,dx_t. \tag{3}$$

By merging constant terms in the transition into the normalization factor, the transition step can be modeled as

$$\hat{p}_\theta(x_{t-1}|x_t) = p_\theta(x_{t-1}|x_t)\,r(x_{t-1}, y). \tag{4}$$

The proof is given in the Appendix material. Hence, rather than considering a Gaussian posterior, as in DPS Chung et al. (2023b), any distance or loss function can be used. A similar idea was suggested in Steered Diffusion Nair et al. (2023). Another valid transition step of the perturbed process is

$$\hat{p}_\theta(x_{t-1}|x_t) = p_\theta(x_{t-1}|x_t)\frac{r(x_{t-1}, y)}{r(x_t, y)}, \tag{5}$$

which adopts the notion of reciprocal distance from the previous timestep. We establish the relationship between the pertrubed latent distribution and the distance functions, which are revisited in Section 3.3

### 3.2 INFERENCE-TIME GUIDANCE OF DIFFUSION MODELS

The same formulation can also be viewed in terms of transition probabilities. Consider a pretrained unconditional diffusion model on a specific domain. The problem at hand is to guide the diffusion model during inference time conditioned on $y$ (as defined earlier). Dhariwal et al. Dhariwal & Nichol (2021) proposed a general strategy to perform this by conditioning on $y$ and finding the resultant marginal distribution

$$p(x_{t-1}|x_t, y) = p(x_{t-1}|x_t)\,p(y|x_{t-1}), \tag{6}$$

where $x_t$ is the latent from the previous timestep.

By assuming the distribution $p(y|x_{t-1})$ has much lower curvature compared to $p(x_{t-1}|x_t)$, and considering the marginal distribution close to $x_{t-1}$,

$$\log p(y|x_{t-1}) = (x_{t-1} - \mu)\nabla_{x_{t-1}} \log p(y|x_{t-1}), \tag{7}$$

$$g = \nabla_{x_{t-1}} \log p(y|x_{t-1}),$$

where $\mu$ denotes the Gaussian mean in the kernel and $g$ is the conditional gradient.

Plugging back into $\log(p(x_{t-1}|x_t, y))$,

$$\log(p(x_{t-1}|x_t, y)) = (x_{t-1} - \mu - \Sigma g)^T \Sigma^{-1} (x_{t-1} - \mu - \Sigma g) + C, \tag{8}$$
$$p(x_{t-1}|x_t, y) \sim \mathcal{N}(\mu + \Sigma g, \Sigma),$$

where $\Sigma$ denotes the covariance matrix of the Gaussian kernel and $C$ is a constant.

Hence, the reverse sampling equation becomes

$$x_{t-1} = \frac{1}{\sqrt{\alpha_t}} \left( x_t - \frac{1 - \alpha_t}{\sqrt{1 - \bar{\alpha}_t}} \epsilon_\theta(x_t) \right) + \sigma_t \epsilon + \Sigma \frac{dr(x_{t-1}, y)}{dx_{t-1}}, \quad \epsilon \sim \mathcal{N}(0, I), \tag{9}$$

where $\alpha_t$ is the noise schedule, $\bar{\alpha}_t = \prod_{i=1}^{t} \alpha_i$ the cumulative variance schedule, $\epsilon_\theta(x_t)$ the noise estimate from the U-Net, and $\sigma_t^2$ the variance of the diffusion kernel (so $\Sigma = \sigma_t^2 I$).

### 3.3 SHORTCOMINGS OF EXISTING METHODS

Although the energy-based guidance theory supports guidance as a function of the current latent estimate, almost all loss-based guidance techniques derive the distance function as a function of $x_t$ rather than $x_{t-1}$ (both already defined) and compute the gradient based on the previous sample. Although this approach works for many tasks, it requires backpropagating through the neural network and modeling the score function for the guidance correction term. This limits the use of classifier guidance since existing diffusion architectures that produce photorealistic results are often very bulky. One can see why the existing framework that utilizes the derivative with respect to the previous sample works by taking a closer look at Equation (5). As we can see, a reciprocal distance over the previous timestep latent $x_t$ is a valid distance guidance function. In the next section, we elaborate on DiffuseGuide.

## 4 PROPOSED METHOD

As mentioned in the previous section, existing works utilize the derivative with respect to the previous step for guidance; one reason is to use an off-the-shelf auxiliary distance function on the MMSE estimate at each step $\hat{x}_t$, which enables the use of general image-space functions for guidance. Here, the MMSE estimate is defined as

$$\hat{x}_t = \frac{x_t - \sqrt{1 - \bar{\alpha}_t} \epsilon_\theta(x_t)}{\sqrt{\bar{\alpha}_t}}, \tag{10}$$

where $\epsilon_\theta(x_t)$ is the noise estimate and $\bar{\alpha}_t$ the cumulative variance schedule.

Another observation is that finding the derivative with respect to the current step requires computing $\hat{x}_{t-1}$, which again requires an additional forward pass through the diffusion network. Hence, the dilemma of backpropagating through the U-Net for guidance remains unresolved.

We found a simple yet effective solution: if we look at the ODE estimate at each step proposed by Song et al. Song et al. (2021a), in the extreme case of deterministic sampling, the next step can be decomposed as

$$x_{t-1} = \sqrt{\alpha_{t-1}} \hat{x}_t + \sqrt{1 - \alpha_{t-1}} \epsilon_\theta(x_t). \tag{11}$$

### 4.1 DOUBLE-DESCENT CLASSIFIER GUIDANCE

Rather than perturbing the Gaussian kernel at each timestep, we perturb the components $\hat{x}_t$ and $\epsilon_\theta(x_t)$ by a small amount. Specifically, we perform:

$$\hat{x}_t = \hat{x}_t - c \, \sigma_t^2 \frac{\partial r(\hat{x}_t, y)}{\partial \hat{x}_t} \tag{12}$$

$$\epsilon_\theta(x_t) = \epsilon_\theta(x_t) - d \, \sigma_t^2 \frac{\partial r(\hat{x}_t, y)}{\partial \epsilon_\theta(x_t)}$$

$$x_{t-1} = \frac{1}{\sqrt{\alpha_t}} \left( x_t - \frac{1 - \alpha_t}{\sqrt{1 - \bar{\alpha}_t}} \epsilon_\theta(x_t) \right) + \sigma_t \epsilon + c_t \, \sigma_t^2 \frac{\partial r(\hat{x}_t, y)}{\partial \hat{x}_t} + d_t \, \sigma_t^2 \frac{\partial r(\hat{x}_t, y)}{\partial \epsilon_\theta(x_t)},$$

where $c$ and $d$ are scalar hyperparameters, $c_t, d_t$ are timestep-dependent scaling factors, and $\sigma_t^2$ is the diffusion variance at step $t$.

We perform a *double descent*: descent on $\hat{x}_t$ guides effectively at the end of diffusion where $\alpha_{t-1}$ is close to one, and descent on $\epsilon_\theta(x_t)$ is most effective in early steps. During this descent, we treat the optimization problem like half-quadratic splitting Zhang et al. (2021). Since $\hat{x}_t$ and $\epsilon(x_t)$ are orthogonal at any step, the maximal component of the shift in $x_{t-1}$ due to guidance on $\hat{x}_t$ occurs through $\hat{x}_t$. Hence, we define

$$c_t = -c\sqrt{\alpha_{t-1}}. \tag{13}$$

Similarly, we define $d_t$ as the maximal component of $\epsilon_\theta(x_t)$ in $x_{t-1}$:

$$d_t = d\,\frac{1-\alpha_t}{\sqrt{\alpha_t}\sqrt{1-\bar{\alpha}_t}}. \tag{14}$$

This provides effective guidance at all timesteps, unlike MGD He et al. (2023), which primarily guides later timesteps. In the following section, we propose an effective empirical estimate for $c$ and $d$ that works for a wide range of tasks.

## 4.2 A Gradient-Dependent Scaling-Factor Estimate

Distance-over-Gradients (DOG) Ivgi et al. (2023) was proposed as an effective parameter-free dynamic step-size schedule for SGD problems. According to DOG, given any stochastic gradient descent optimization problem, the distance over the gradient works as an effective learning rate. Recent works Wu et al. (2023b) interpret diffusion sampling as a stochastic optimization problem. Inspired by both, we adopt an empirical guidance estimate of the form:

$$\gamma_t = \begin{cases} \frac{1e^{-5}}{\sqrt{\tilde{g}_T^2}}, & \text{if } t = T \\ \frac{\max_{i>t}|f_i - f_T|}{\sqrt{\sum_{i=t}^T \tilde{g}_i^2}}, & \text{otherwise}, \end{cases} \tag{15}$$

where $\tilde{g}_t = \nabla_{f_t}\mathcal{L}(f_t, y)$ is the gradient of the chosen loss function with respect to $f_t$ at timestep $t$, $f_t \in \{\hat{x}_t, x_t, \epsilon_\theta(x_t)\}$, and $f_T$ is the terminal value at the last timestep.

We observed that this empirical estimate works well for first-order sampling involving DPS Chung et al. (2023b) as well. Using Equation (15), we estimate $c$ and $d$ accordingly by substituting $f_i$ as $\hat{x}_t$ and $\epsilon_\theta(x_t)$.

## 4.3 Differentiable-Augmentation Classifier Guidance

A common practice when performing classifier guidance is to use the noisy estimate at timestep $t$ and compute a loss to regularize the current prediction. However, in many cases, such guidance can produce artifacts and color shifts (see Figure 3 and Figure 5) due to excessive or insufficient guidance at intermediate timesteps that pushes samples off-manifold. An effective solution is to imitate different artifact/color-shift variants on both the source and the target and use these augmented versions to stabilize guidance. We introduce *DiffuseAugment*, an augmentation strategy for diffusion guidance during inference. Given an intermediate sample $x_t$ and condition $y$, we augment $\hat{x}_t$ and $y$ with differentiable augmentations:

$$\hat{x}_t^{aug}, y^{aug} = T(\hat{x}_t, y), \tag{16}$$

where $T(\cdot)$ is a differentiable augmentation operator including random cutouts, random translations, and color saturation. The augmentation of $y$ depends on the input signal: for label-based conditioning (e.g., identity or text), we do not augment $y$; for image-space conditioning, we apply the same random augmentation to $y$ as to $x_t$. We average the loss across augmentations. We find that DiffuseAugment significantly boosts fidelity and sampling quality. Results are presented in Section 6.

## 5 Experiments

Because our method applies to both linear and non-linear inverse tasks, for linear tasks we follow DPS and evaluate on two benchmarks: (1) ImageNet Deng et al. (2009) and (2) CelebA Liu et al. (2015).

| Method | Inpaint (Box) | | | | Colorization | | | | SR ($\times 4$) | | | | Gaussian Deblur | | | |
|---|---|---|---|---|---|---|---|---|---|---|---|---|---|---|---|---|
| | PSNR ↑ | SSIM ↑ | LPIPS ↓ | FID ↓ | Cons ↑ | SSIM ↑ | LPIPS ↓ | FID ↓ | PSNR ↑ | SSIM ↑ | LPIPS ↓ | FID ↓ | PSNR ↑ | SSIM ↑ | LPIPS ↓ | FID ↓ |
| Score-SDE Song et al. (2021b) | 9.57 | 0.329 | 0.634 | 94.33 | 0.1627 | 0.3996 | 0.6609 | 118.86 | 20.75 | 0.5844 | 0.3851 | 53.22 | 23.39 | 0.632 | 0.361 | 66.81 |
| ILVR Choi et al. (2021) | - | - | - | - | - | - | - | - | 26.14 | 0.7403 | 0.2776 | 52.82 | - | - | - | - |
| DPS Chung et al. (2023a) | 19.39 | 0.610 | 0.3766 | 58.89 | 0.0069 | 0.5404 | 0.5594 | 55.61 | 17.36 | 0.4969 | 0.4613 | 56.08 | 20.52 | 0.5824 | 0.3756 | 52.64 |
| MGD He et al. (2023) | 27.21 | 0.7460 | 0.2197 | 11.83 | 0.0018 | 0.6865 | 0.4549 | 38.22 | 27.51 | 0.7852 | 0.2464 | 60.21 | 27.23 | **0.7695** | 0.2327 | 51.59 |
| Ours | **28.84** | **0.8491** | **0.1432** | **5.96** | **0.0014** | **0.7775** | **0.3036** | **20.89** | **29.47** | **0.8429** | **0.1757** | **46.95** | **27.30** | 0.7672 | **0.2202** | **42.70** |

Table 2: Quantitative evaluation of image restoration tasks on CelebA 256×256-1k with $\sigma_y = 0.05$, We utilize 100 inference steps for all methods

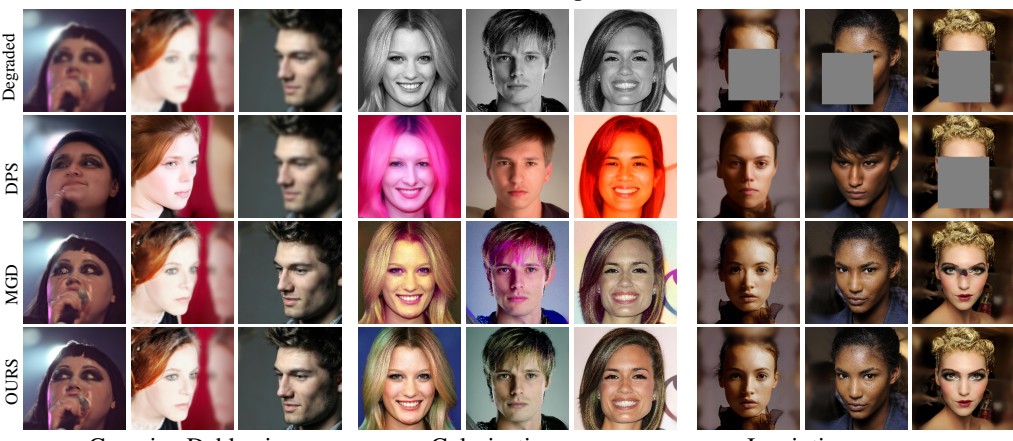

Gaussian Deblurring          Colorization          Inpainting

Figure 3: Qualitative comparisons for Linear Tasks on CelebA dataset for 100 inference steps

| Method | Inpaint (Box) | | | | Colorization | | | | SR ($\times 4$) | | | | Gaussian Deblur | | | |
|---|---|---|---|---|---|---|---|---|---|---|---|---|---|---|---|---|
| | PSNR ↑ | SSIM ↑ | LPIPS ↓ | FID ↓ | Cons ↑ | SSIM ↑ | LPIPS ↓ | FID ↓ | PSNR ↑ | SSIM ↑ | LPIPS ↓ | FID ↓ | PSNR ↑ | SSIM ↑ | LPIPS ↓ | FID ↓ |
| Score-SDE Song et al. (2021b) | 9.66 | 0.2087 | 0.7375 | 133.54 | 0.1723 | 0.3105 | 0.8197 | 194.87 | 14.07 | 0.2468 | 0.6766 | 129.91 | 15.39 | 0.3158 | 0.620 | 134.67 |
| ILVR Choi et al. (2021) | - | - | - | - | - | - | - | - | 15.51 | 0.4033 | 0.5253 | 64.13 | - | - | - | - |
| DPS Chung et al. (2023a) | 15.23 | 0.4261 | 0.6087 | 97.90 | 0.021 | 0.3774 | 0.8011 | 106.25 | 14.94 | 0.3258 | 0.6594 | 87.26 | 17.19 | 0.3980 | 0.5817 | 84.74 |
| MGD He et al. (2023) | 21.94 | 0.6920 | 0.2410 | 40.30 | 0.0057 | 0.5809 | 0.5427 | 73.75 | 23.12 | 0.6025 | 0.3936 | 70.83 | 23.13 | 0.6092 | 0.3695 | 61.49 |
| Ours | **23.49** | **0.7271** | **0.2001** | **30.72** | **0.0055** | **0.6804** | **0.3362** | **52.76** | **24.23** | **0.6818** | **0.2884** | **43.00** | **23.31** | **0.6157** | **0.3566** | 58.38 |

Table 3: Quantitative evaluation of image restoration tasks on ImageNet 256×256-1k with $\sigma_y = 0.05$. **Bold**: best, We utilize 100 inference steps for all methods

For non-linear tasks, we follow Freedom and evaluate using the CelebA dataset. For linear tasks, we evaluate super-resolution ($\times 4$), colorization, inpainting (box), and Gaussian deblurring. For non-linear tasks, we evaluate face sketch guidance, face parse-map guidance, and face ID guidance. Since our method is loss-guided, we compare against existing loss-guided sampling methods. Although we acknowledge the parallel line of work on inverse problems without backpropagation Wang et al. (2023); Kawar et al. (2021), we exclude those methods since they tackle only linear inverse problems, whereas loss-guided models are generic.

**Implementation Details:** We perform all experiments on NVIDIA A6000 GPUs. For ImageNet tasks, we utilize the unconditional model released by Guided Diffusion. For linear face tasks, we use the model trained on the FFHQ dataset Karras et al. (2017) and evaluate on CelebA Liu et al. (2015), as in DPS. For non-linear tasks, we follow Freedom and utilize the unconditional model trained on CelebA. We evaluate using conditions derived from existing networks. For the high-resolution results in Figure 2, we use the class-conditional $512 \times 512$ model released by Guided Diffusion. Unless stated otherwise, we use 100 sampling steps. For style transfer, we utilize Stable Diffusion v1.5 Rombach et al. (2021). Our sampling method is generic and compatible with different samplers; in our experiments, we rescaled the DDPM schedule. We fix the number of DiffuseAugment augmentations to 8. For all non-linear tasks, following MGD He et al. (2023), we utilize three time-travel sampling Lugmayr et al. (2022) steps. Even with time-travel sampling, our overall compute time is comparable to Freedom due to savings from bypassing backpropagation through the U-Net (see Appendix). We compare against Freedom (first-order; requires backprop) and MGD. For non-linear tasks, we additionally apply gradient clipping in $(-0.2, 0.2)$. For evaluations, we use MGD, DPS, Score SDE and ILVR. The MGD implementation follows parameters from the original paper (guidance scale 100 for all linear tasks with manifold projection). We detail more on the benchmarks and the guidance functions used in the appendix.

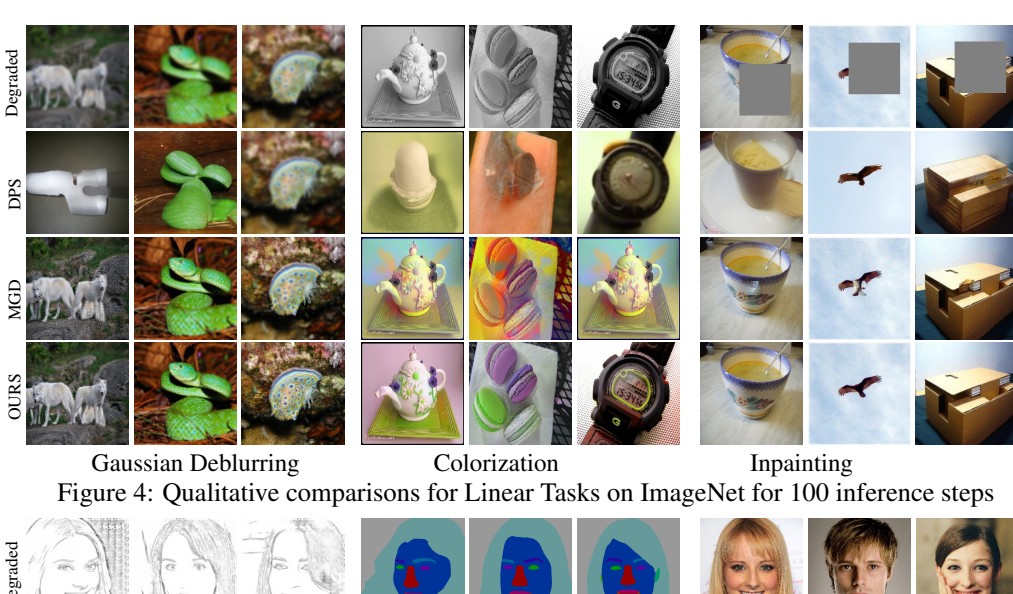

Gaussian Deblurring      Colorization      Inpainting

Figure 4: Qualitative comparisons for Linear Tasks on ImageNet for 100 inference steps

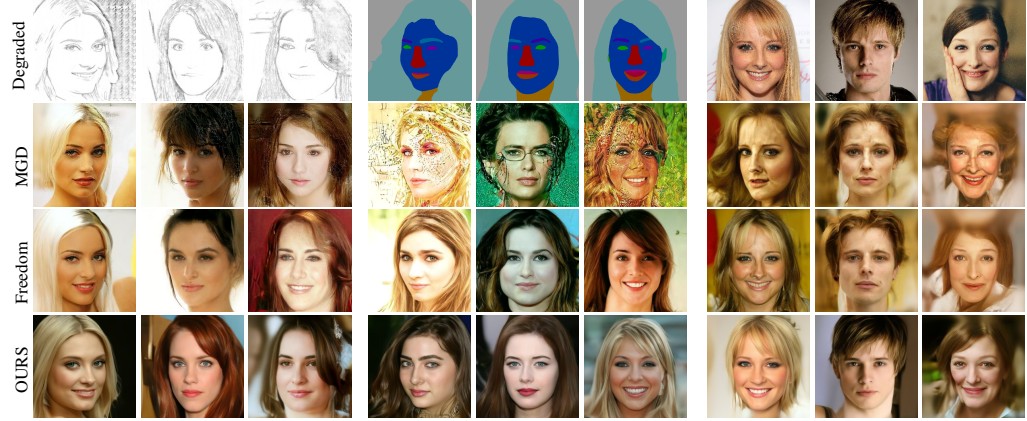

Face-sketch guidance     Face-parse guidance     ID guidance

Figure 5: Qualitative comparisons for Non-linear Tasks on CelebA dataset for 100 inference steps

**Qualitative Analysis:** Qualitative face results are given in Figure 3: Gaussian deblurring, super-resolution, and colorization. DPS struggles with 100 diffusion steps because its scaling factor is not strong enough to provide proper guidance within that budget (posterior noise set to 0.05 in all experiments). MGD works well for deblurring and inpainting but fails for colorization, which requires early guidance for natural color flow. In contrast, our method produces more natural images across all cases due to guidance throughout all timesteps. Since restorative face tasks are easier (limited domain), we show ImageNet results in Figure 4. On ImageNet, DPS performance drops more because the problem is more ill-posed (e.g., the eagle example). Our method produces more realistic images, especially in colorization, again due to early gradient flow. For fairness, face models are trained on FFHQ Karras et al. (2017) and tested on CelebA Liu et al. (2015); ImageNet experiments use the validation set. As in Appendix B, we evaluate on CelebA and ImageNet. Results for face restoration are shown in Table 2 and Table 3. SDEdit Meng et al. (2021) fails on face inpainting and colorization because a single perturbation in the noisy domain can push images off-manifold. DPS requires more steps for proper guidance. ILVR is designed for super-resolution; thus we evaluate it only for that task. DPS and MGD apply to all cases. Our approach yields better results than baselines due to guided gradient flow, improving reconstruction quality. On faces, the improvement is most pronounced for colorization (e.g., an 18 FID-point boost over the baseline). ImageNet linear inverse problems are more complex than faces, so overall metrics are lower.

**Analysis on Non-Linear Tasks:** We compare the performance against Freedom Yu et al. (2023) and MGD He et al. (2023). Figure 5 shows qualitative results. Freedom produces realistic outputs even for the challenging parse-map–to–face task, likely because backpropagating through the U-Net purifies gradient flow. With DiffuseAugment, our gradients are likewise purified, yielding realistic results. MGD does not produce realistic outputs for sketch–to–image and anime–to–face synthesis. We evaluate *Distance* (the $L_2$ norm between generated and original degradation maps), LPIPS, and FID. Note that artifacts in MGD are not always fully reflected by these metrics. The corresponding

| Method | Semantic Parsing | | | ID Guidance | | | Face Sketch | | |
|---|---|---|---|---|---|---|---|---|---|
| | Distance↓ | LPIPS↓ | FID↓ | Distance↓ | LPIPS↓ | FID↓ | Distance↓ | LPIPS↓ | FID↓ |
| | *First-order* | | | | | | | | |
| Freedom Yu et al. (2023) | 1864.51 | 0.6030 | 66.89 | 0.3767 | 0.7058 | 81.40 | 39.05 | 0.6583 | 86.51 |
| | *Zeroth-order* | | | | | | | | |
| MGD He et al. (2023) | **2698.27** | 0.6995 | 104.32 | 0.4291 | 0.7178 | 92.61 | 39.34 | 0.6576 | 70.42 |
| Ours | 2722.51 | **0.6199** | **79.42** | **0.3780** | **0.5932** | **82.70** | **39.03** | **0.5509** | **69.51** |

Table 4: Non-linear tasks. Best results out of zeroth-order optimization algorithms are highlighted.

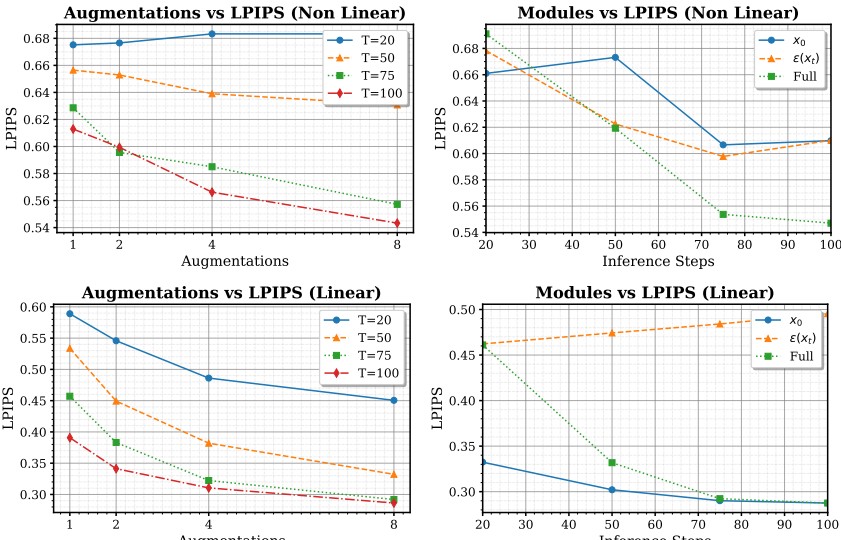

Figure 6: Ablation analysis: **Top Row:** FaceID guidance **Bottom Row:** ImageNet superresolution metrics are in Section 5. Compared to first-order methods, we obtain better FID and LPIPS across all cases, with significantly lower compute than Freedom.

## 6 ABLATION STUDIES

We perform extensive ablations on the effect of DiffuseAugment and on each guidance term. For ablations, we use 100 images and report average LPIPS due to the volume of experiments.

**Effect of DiffuseAugment:** We vary the number of augmentations for both linear and non-linear tasks. For the linear task, we choose ImageNet super-resolution ($\times 4$). We vary diffusion steps $T \in \{20, 50, 75, 100\}$ and report average LPIPS. For linear tasks, even with low $T$ (e.g., $T = 20$), increasing augmentations to 8 markedly improves perceptual quality, matching $T = 50$ with only two augmentations. For non-linear tasks, the benefit is muted or negative at $T = 20$ because most $\hat{x}_t$ remain too noisy for the guidance network (e.g., ArcFace Deng et al. (2019)), which yields irregular gradients. As $T$ increases and gradients stabilize, DiffuseAugment yields substantial gains.

**Effect of Different Guidance Components:** Figure 6 ablates different terms (DiffuseAugment fixed at 1; time-travel sampling off). Guiding with $\hat{x}_t$ alone shows a performance dip for small $T$; early guidance through $\hat{x}_t$ is weak, and time-travel sampling (if used) would require careful tuning. Guiding via the output noise $\epsilon_t$ helps when $T$ is small, but the effect diminishes as $T$ increases. Our double-descent guidance provides both early and late guidance, improving perceptual quality—especially for non-linear inverse problems where gradient estimates are noisier. For linear inverse problems, double descent offers smaller gains and can slightly degrade performance in some settings (see Figure 6). Additional examples are in the Appendix.

## 7 CONCLUSION

We proposed an improvement to loss-guided, zero-shot conditional generation with unconditional diffusion models. Specifically, we introduced a sampling technique that removes the need to back-propagate through the diffusion U-Net, enabling guidance for general inverse problems. We also proposed an empirical, automatic scaling function that removes manual tuning of guidance scales and start/end guidance steps. Finally, we introduced a differentiable data-augmentation method that significantly improves sampling fidelity. We demonstrated results across four linear and three non-linear tasks on faces and natural images. Our sampling technique produces photorealistic samples with lower sampling time and higher fidelity than existing methods.

## 8 ETHICS STATEMENT

This work studies generative modeling from a theoretical and methodological perspective. All datasets used (ImageNet,CelebA, FFHQ) are publicly available and widely adopted in research, involving no human subjects or private data. While generative models may be misused to create harmful content, our contributions are intended solely to advance scientific understanding and efficiency of visual generation. We declare no conflicts of interest, and all results are reproducible with the code and checkpoints that will be released.

## 9 REPRODUCIBILITY STATEMENT

We have taken steps to ensure reproducibility of our results. The datasets are publicly available and described in the appendix. Model architecture, training details, and hyperparameters are provided in Section 4 and Appendix. We report all experimental protocols, ablations, and evaluation metrics. Code, pretrained checkpoints, and instructions to reproduce our results will be released upon publication.

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

APPENDIX

## A EVALUATION BENCHMARK DETAILS:

We evaluate four tasks. For inpainting, we choose a random box mask of size $128 \times 128$ pixels. For Gaussian deblurring, we apply a $61 \times 61$ Gaussian blur with kernel intensity 3.0. For super-resolution, we downsample images using bicubic downsampling to $64 \times 64$. We report PSNR and SSIM for restoration quality, LPIPS Zhang et al. (2018) for perceptual similarity, and FID Seitzer (2020) for distributional similarity. For colorization, we also report a consistency measure: the MSE between grayscale versions of the reconstruction and the original image (lower is better). For non-linear inverse functions, we evaluate three tasks: face parse guidance, face ID guidance, and face sketch guidance. For sketches, we utilize an open-source pretrained network that converts faces to sketches Xiang et al. (2022). We also evaluate Score-SDE Meng et al. (2021). For super-resolution we additionally evaluate the zero-shot method ILVR Choi et al. (2021).

## B GUIDANCE FUNCTIONS UTILIZED:

For colorization, we convert to YCbCr and take the Y component as the measurement. We compare two loss-guided diffusion methods: DPS Chung et al. (2023b) and MGD He et al. (2023). We use 1000 images from CelebA Liu et al. (2015) as in Freedom Yu et al. (2023) and obtain the corresponding non-linear map for each image . As the guidance function for sketches, we use the Euclidean distance between the generated sketch from $\hat{x}_t$ and the target sketch. For face parse guidance, we utilize BiSeNet Yu et al. (2018) to derive parse maps from $\hat{x}_t$ and use the Euclidean distance between predicted and ground-truth labels. For face ID guidance, we use Deng et al. (2019) to obtain face embeddings and measure cosine-similarity loss.

## C ALGORITHM OF DIFFUSEGUIDER

We present the over algorithm of dreamguider without time travel sampling and the parameter estimation algorithm in Algorithm 1

## D PROOF FOR PERTURBED MARKOVIAN KERNEL EQUATION

In the main paper, we emphasized that any positive distance function can be utilized for performing conditional generation using the perturbed Markovian kernel equation. hHre we proceed to derive the perturbed transition step. For the proof we closely follow the work from Dickenson et al Sohl-Dickstein et al. (2015). Given a unconditional transition distribution $p_\theta(x_{t-1}|x_t)$ and a distance function $r(.,y)$, where y is the condition provided Please note that we assume $r(.,y)$ has relatively small variance compared to $p_\theta(x_{t-1}|x_t)$, We know that at equilibrium state, the distribution at any timestep $t$ ina diffusion model can be written as

$$p(x_{t-1}) = \int p(x_t)p_\theta(x_{t-1}|x_t)dx_t. \tag{17}$$

To estimate a perturbed transition kernel $\hat{p}(x_{t-1}|x_t)$,we start the perturbed distribution as

$$p(x_{t-1})r(x_{t-1},y) = \int r(x_t,y)p(x_t)\hat{p}_\theta(x_{t-1}|x_t)dx_t. \tag{18}$$

By simple algebraic manipulations, taking $r(x_{t-1},y)$ to the other side, we get

$$p(x_{t-1}) = \int \frac{r(x_t,y)}{r(x_{t-1},y)}p(x_t)\hat{p}_\theta(x_{t-1}|x_t)dx_t. \tag{19}$$

---

**Algorithm 1** Dreamguider

---

**Input:** distance function $r(.,.y)$, condition $y$ , Timesteps $T$
1: $x_T \sim \mathcal{N}(x_T; 0, I)$
2: **for** $t = T - 1, \ldots, 1$ **do**
3:      $\Sigma = \sqrt{1 - \bar{\alpha}_t}$
4:      $\epsilon \sim \mathcal{N}(\epsilon; 0, I)$
5:      $\hat{x}_t = \frac{x_t - \sqrt{1 - \bar{\alpha}_t} \epsilon_\theta(x_t)}{\sqrt{\bar{\alpha}_t}}$
6:      Compute $\frac{dr(\hat{x}_t, y)}{d\hat{x}_t}, \frac{dr(\hat{x}_t, y)}{d\epsilon_\theta(x_t)}$
7:      update $c = ESTIMATE(t, \epsilon_\theta(x_t), \frac{dr(\hat{x}_t, y)}{d\epsilon_\theta(x_t)})$
8:      update $d = ESTIMATE(t, \hat{x}_t, \frac{dr(\hat{x}_t, y)}{d\hat{x}_t})$
9:      $c_t = c\sqrt{\alpha_{t-1}}$
10:      $d_t = -d.\frac{1 - \alpha_t}{\sqrt{\alpha_t}\sqrt{1 - \bar{\alpha}_t}}$
11:      $x_{t-1} = \frac{1}{\sqrt{\alpha_t}}\left(x_t - \frac{1 - \alpha_t}{\sqrt{1 - \bar{\alpha}_t}} \epsilon_\theta(x_t)\right) + \sigma_t \epsilon - c_t \Sigma \frac{dr(\hat{x}_t, y)}{d\hat{x}_t} - d_t \Sigma \frac{dr(\hat{x}_t, y)}{d\epsilon_\theta(x_t)}$
12: **end for**
13: **function** ESTIMATE($t, f_i, g_t$)
14:      **if** $t = T$ **then**
15:          $\gamma_t = \frac{1e^{-5}}{\sqrt{g_T^2}}$
16:          Store $f_T$,
17:      **else**
18:          $\gamma_t = \frac{\max i > t |f_i - f_T|}{\sqrt{\Sigma_{i=i}^T g_t^2}}$
19:      **end if**
20:      Store $\sqrt{\Sigma_{i=i}^T g_t^2}$
21:      **return** $\gamma_t$
22: **end function return** $x_0$

---

By comparing Equation (17) and Equation (19) we can see that one solution for the transitional distribution is

$$\hat{p}_\theta(x_{t-1}|x_t) = p_\theta(x_{t-1}|x_t)\frac{r(x_{t-1}, y)}{r(x_t, y)}. \qquad (20)$$

Also since normalization constants doesn't affect the score function or transition step, Absorbing $x_t$ to the normalization factor of $p_\theta(x_{t-1}|x_t)$, another valid perturbed transition kernel is

$$\hat{p}_\theta(x_{t-1}|x_t) = p_\theta(x_{t-1}|x_t)\frac{r(x_{t-1}, y)}{Z}. \qquad (21)$$

Please note that the term $Z$ does not affect the transition step in the reverse process when the variance of $r(., y)$ is small.

| Method | Freedom | Dreamguider(1) | Dreamguider(2) | Dreamguider(3) |
|---|---|---|---|---|
| Sketch to Face | 24.95 | 17.55 | 27.04 | 35.09 |
| FaceID to Face | 24.94 | 20.45 | 31.89 | 41.80 |
| FaceParse to Face | 56.25 | 48.35 | 75.43 | 107.02 |

Table 5: Non-linear tasks ablation analysis on time taken, the value is represented in seconds

## E TIME COMPARISON FOR DREAMGUIDER WITH TIMETRAVEL SAMPLING AND FREEDOM(FIRST ORDER) FOR NON LINEAR TASKS

We present the time taken by Freedom, a first order algorithm for one step of time travel sampling Lugmayr et al. (2022); Yu et al. (2023) in Table 5

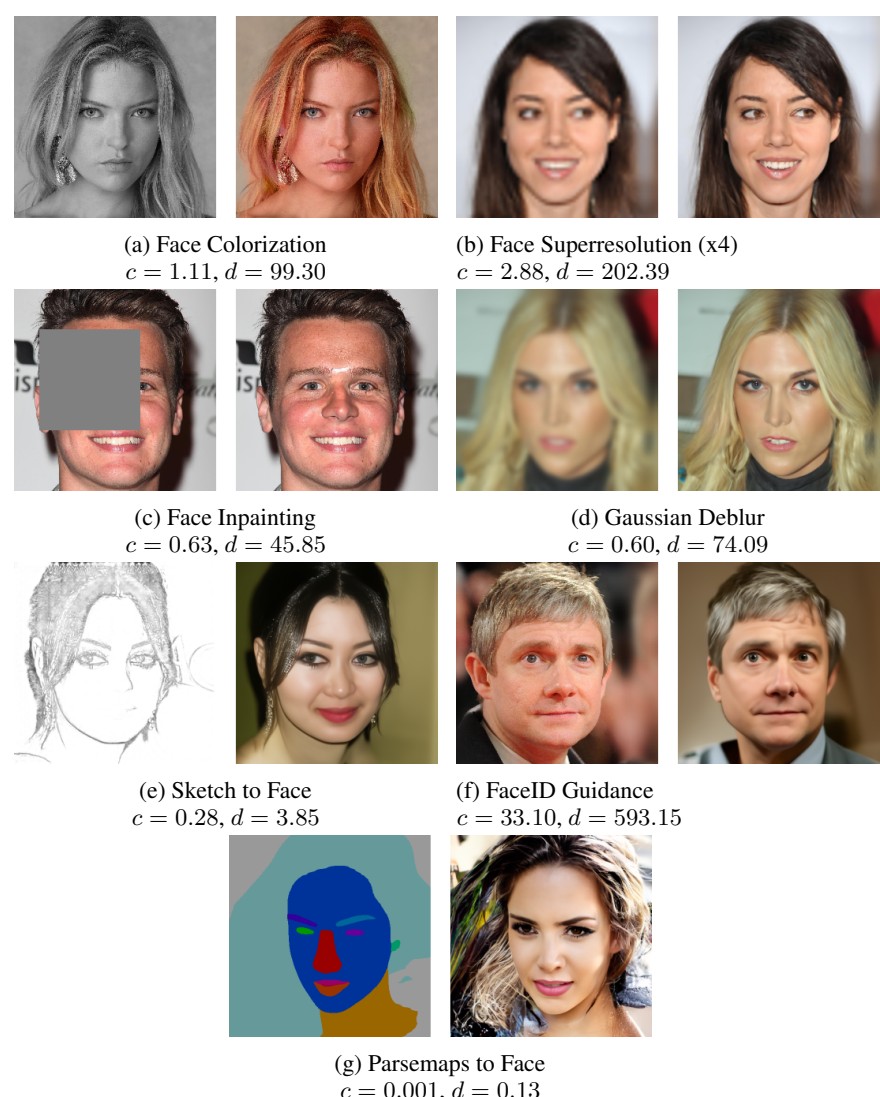

(a) Face Colorization
$c = 1.11, d = 99.30$

(b) Face Superresolution (x4)
$c = 2.88, d = 202.39$

(c) Face Inpainting
$c = 0.63, d = 45.85$

(d) Gaussian Deblur
$c = 0.60, d = 74.09$

(e) Sketch to Face
$c = 0.28, d = 3.85$

(f) FaceID Guidance
$c = 33.10, d = 593.15$

(g) Parsemaps to Face
$c = 0.001, d = 0.13$

Figure 7: Figure illustrating the guidance scales for different tasks.

## F    ESTIMATED PARAMETER VALUE FOR DIFFERENT TASKS

In this section, we present the result and the parameter estimated by our approach for different tasks. For this experiment, we use 100 timesteps of diffusion and present the value at the 100th timestep. Here we define $d$ as the scaling factor of the scaling constant of the the loss derivative relative to $\epsilon_\theta(x_t)$ and c as that of $\hat{x}_t$ as in the main paper . The corresponding results are shown in Figure 7

## G    FUTURE WORK

Although we illustrated the approach across various tasks for pixel-space diffusion models, the direct approach cannot be used for latent diffusion models on linear inverse problems without additional time-travel sampling steps, which increases compute due to VAE reconstruction error. In future work, we will explore optimization strategies to mitigate this. Moreover, while the empirical DOG-based estimate works well across tasks and suggests the existence of an optimal parameter, a thorough mathematical analysis to obtain truly optimal parameters remains open.

## H  LLM Usage

We acknowledge that Large Language Models (LLMs) were used to assist with refining the clarity of the writing in this manuscript.

## I  Non cherry picked results for different tasks.

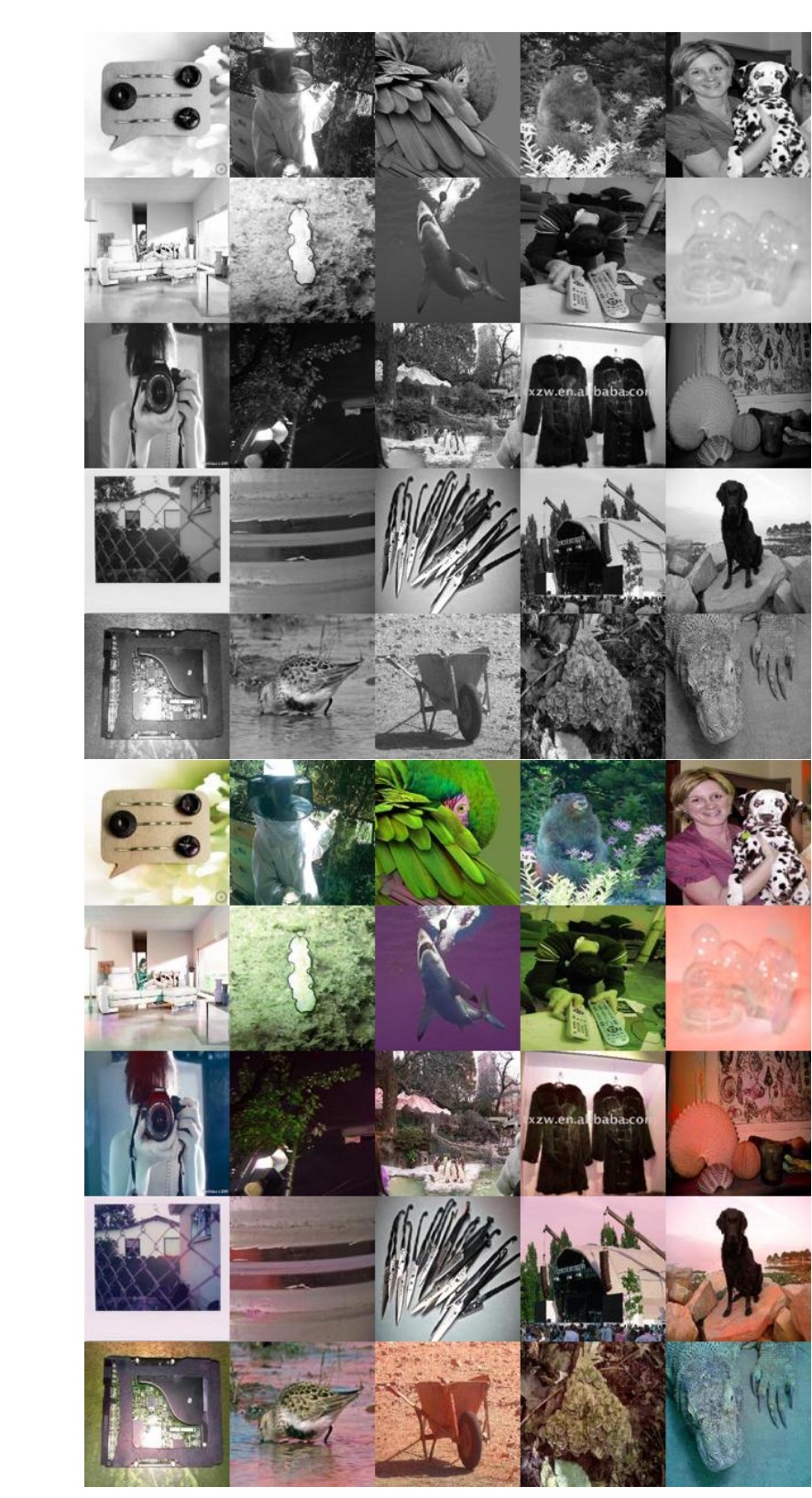

Figure 8: Figure illustrating **Non cherry picked** results for ImageNet colorization

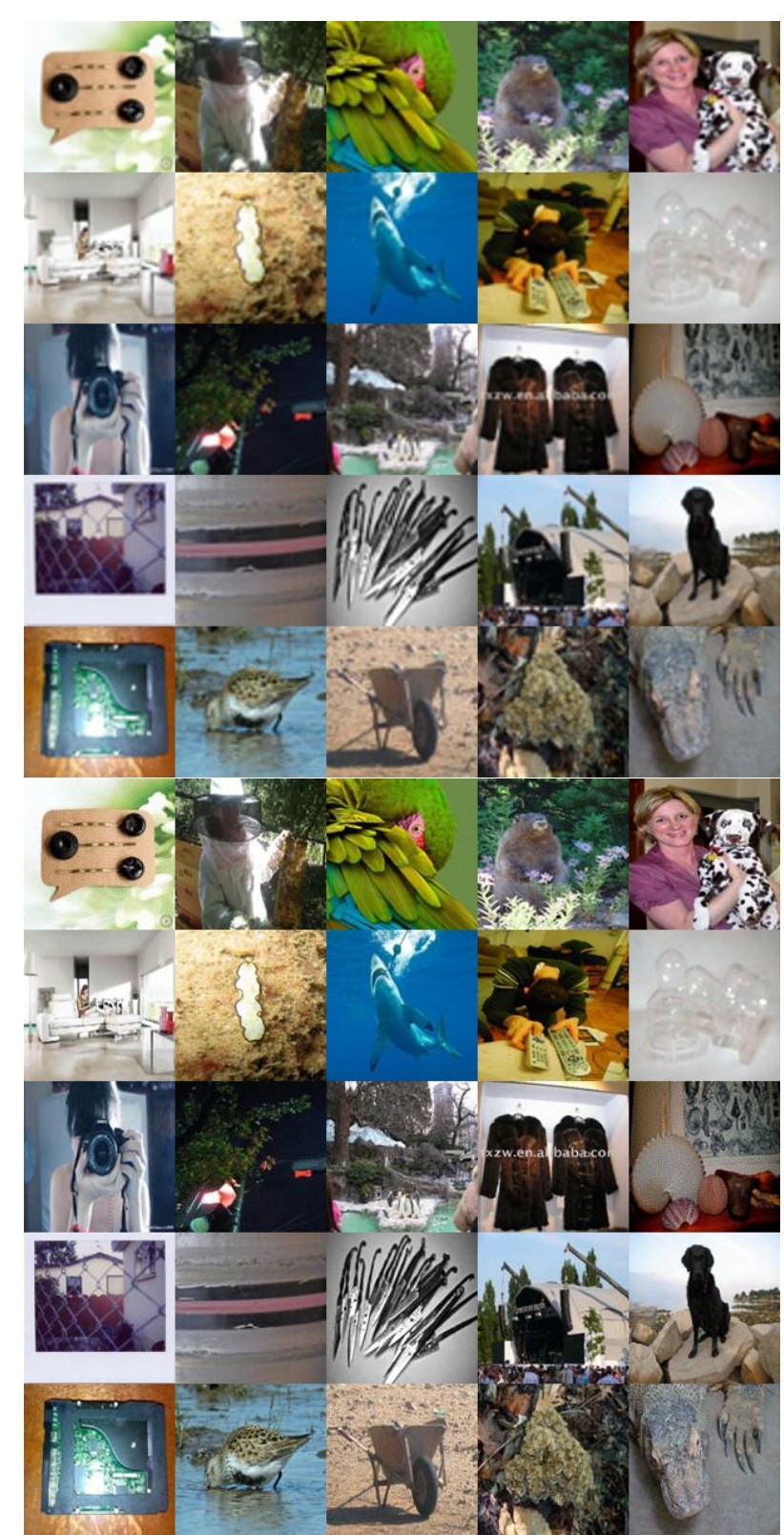

Figure 9: Figure illustrating **Non cherry picked** results for ImageNet superresolution

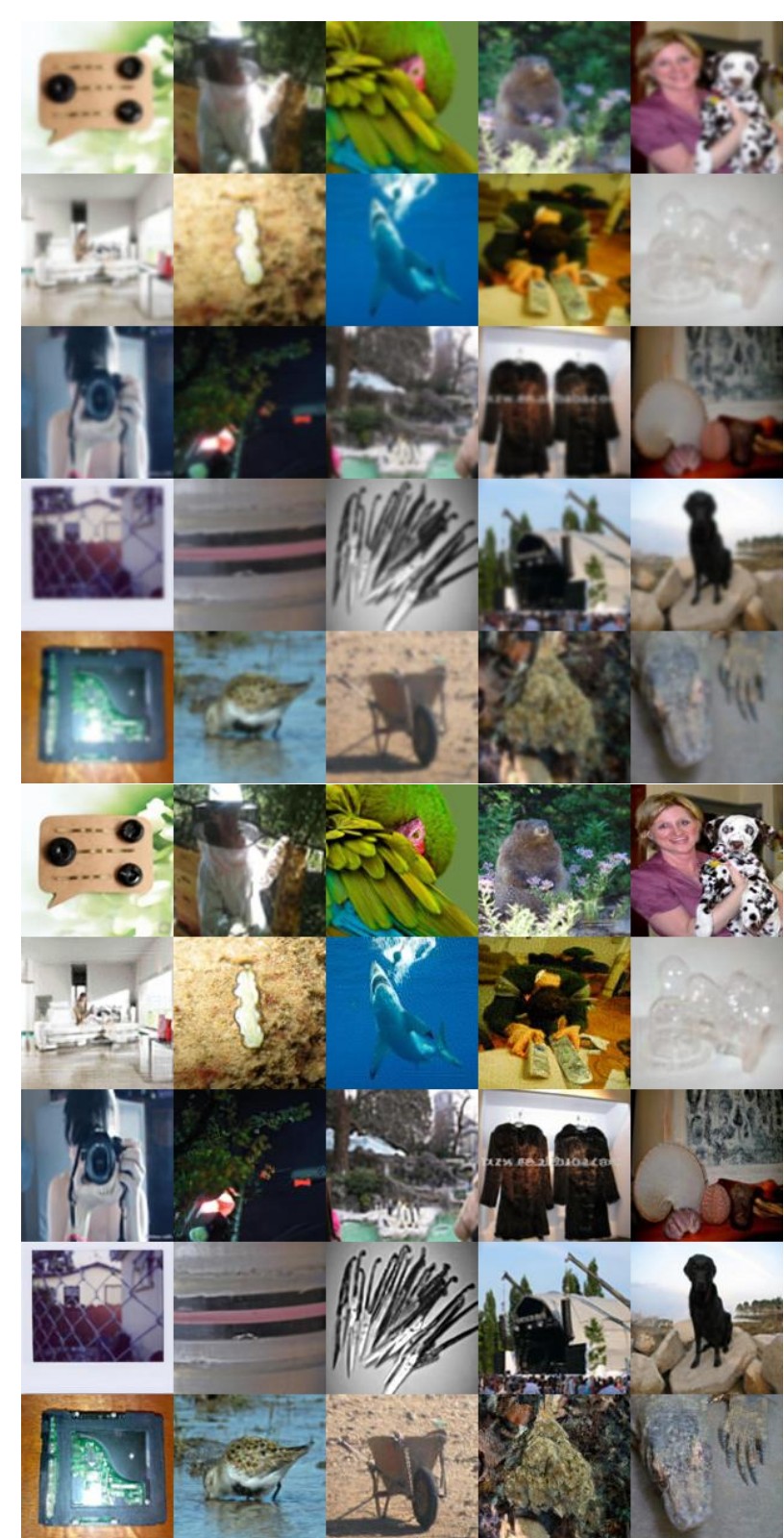

Figure 10: Figure illustrating **Non cherry picked** results for Gaussian deblurring on ImageNet

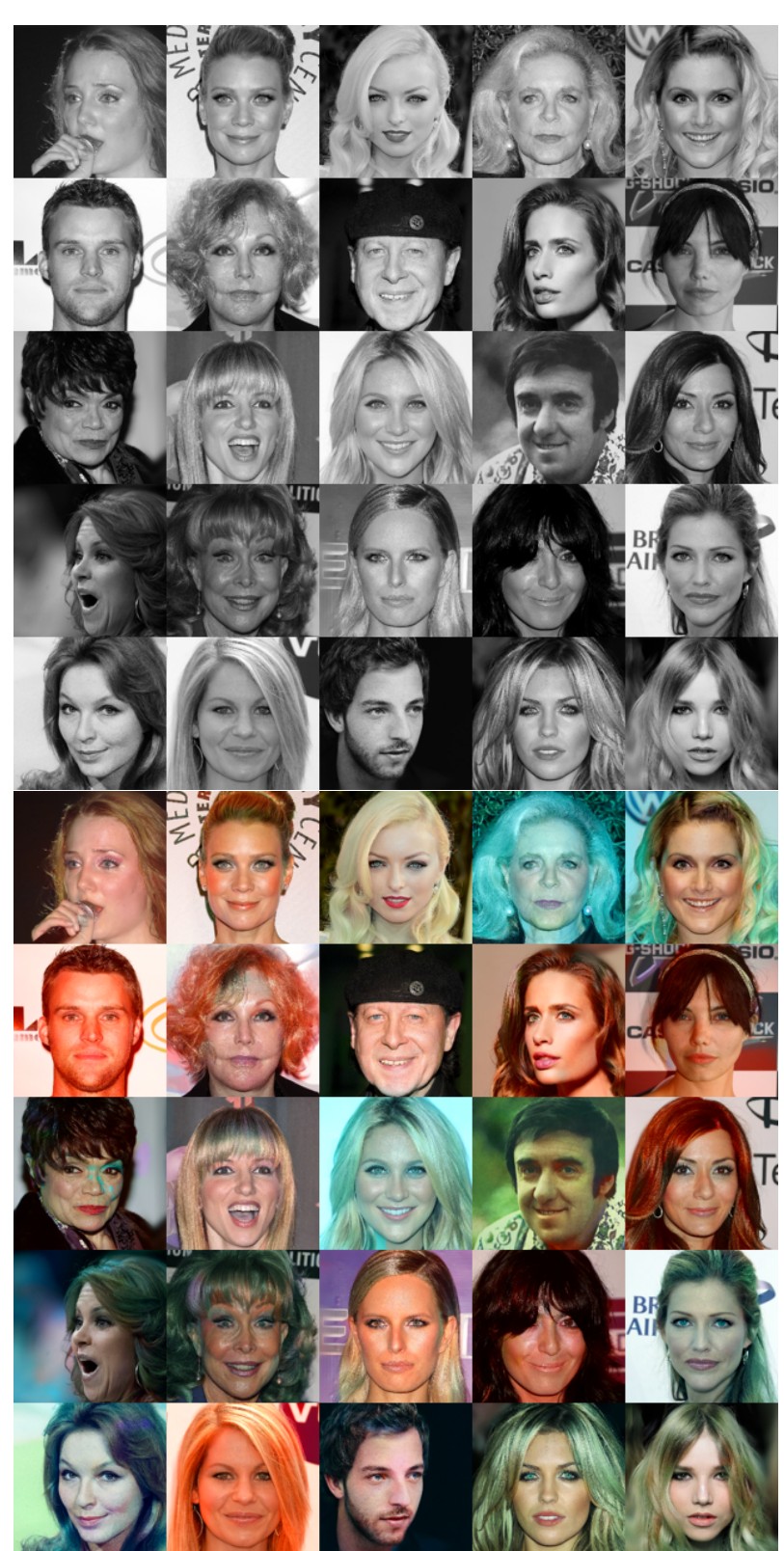

Figure 11: Figure illustrating **Non cherry picked** results for face colorization

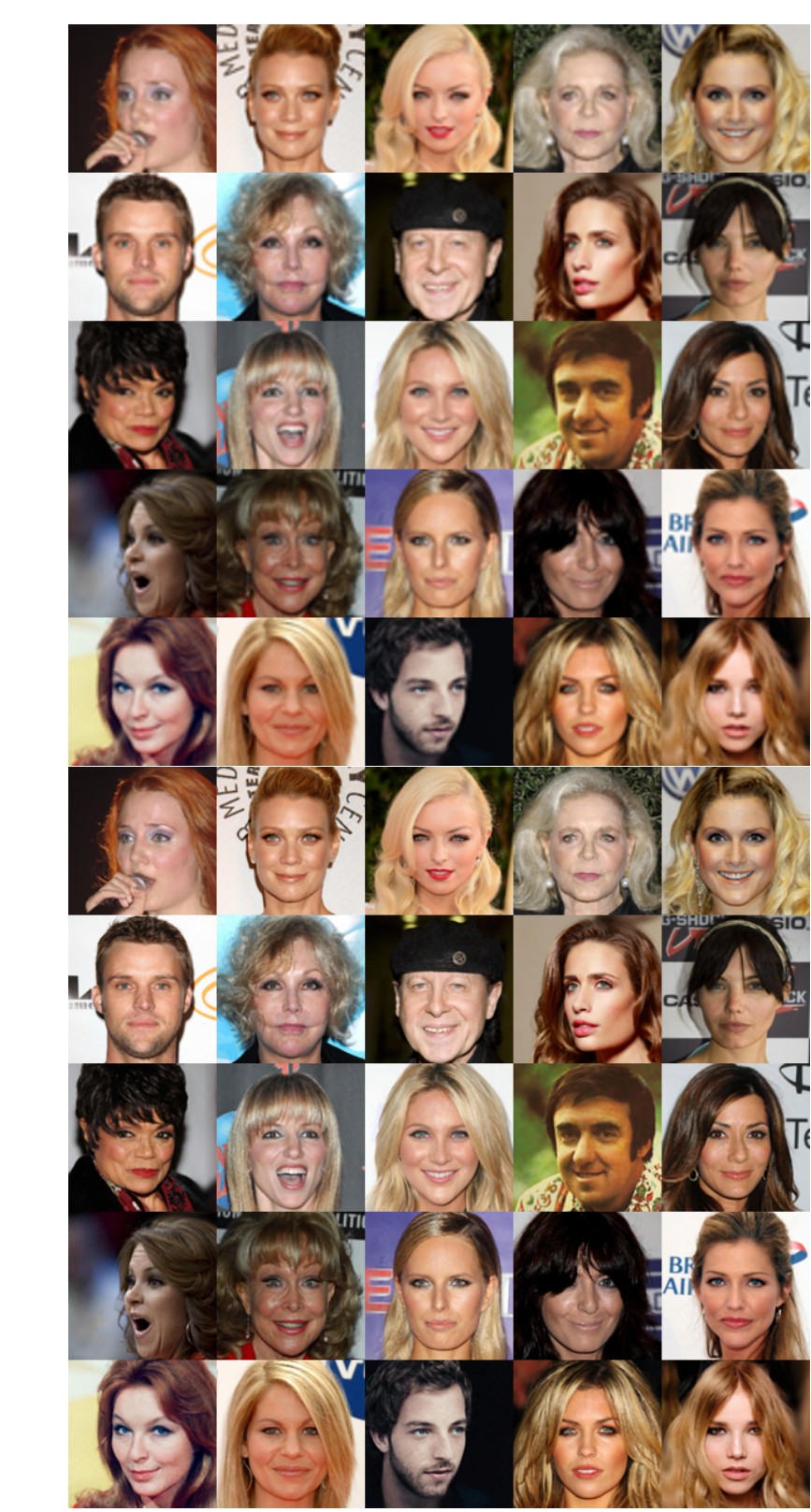

Figure 12: Figure illustrating **Non cherry picked** results for face superresolution

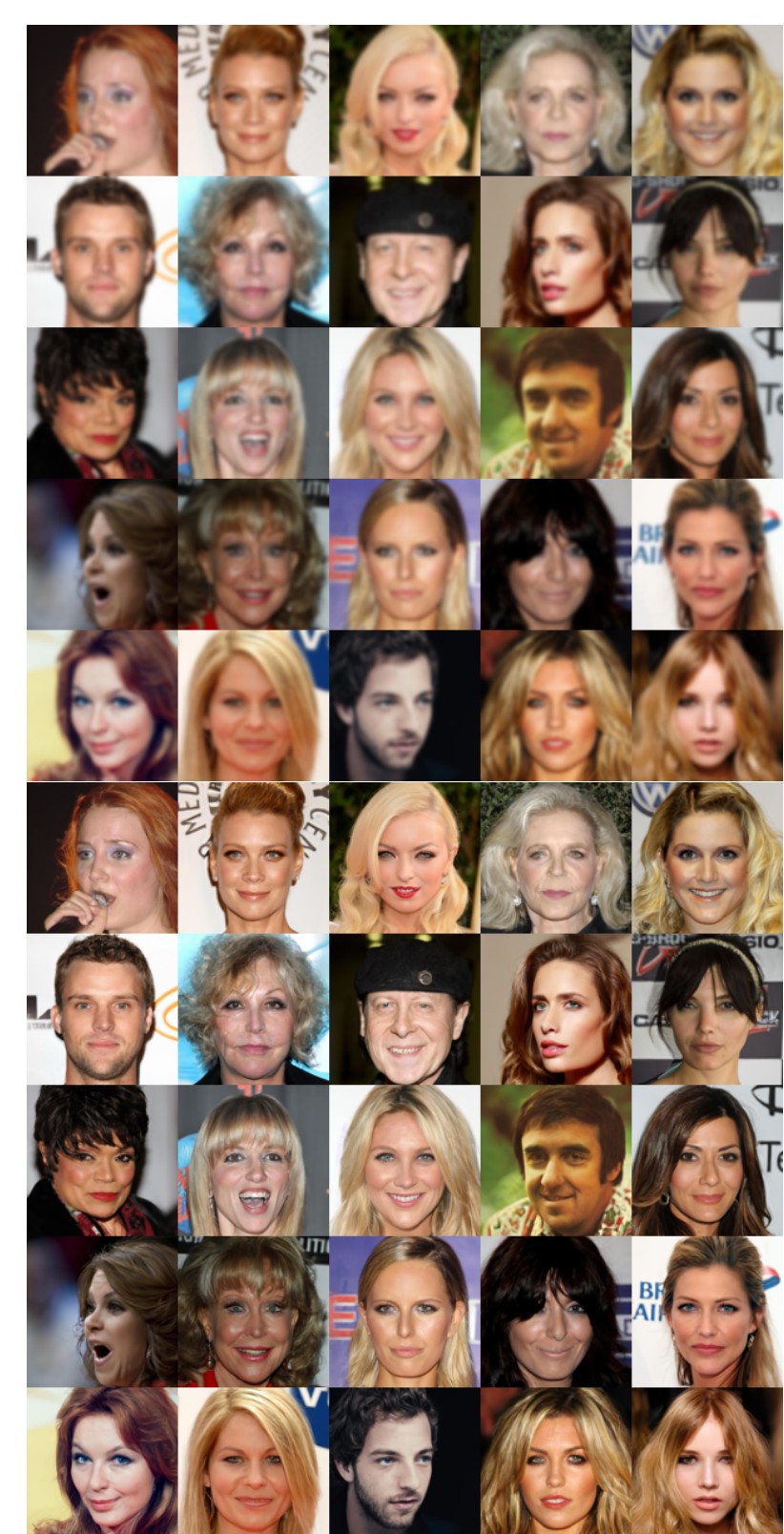

Figure 13: Figure illustrating **Non cherry picked** results for Gaussian Deblurring

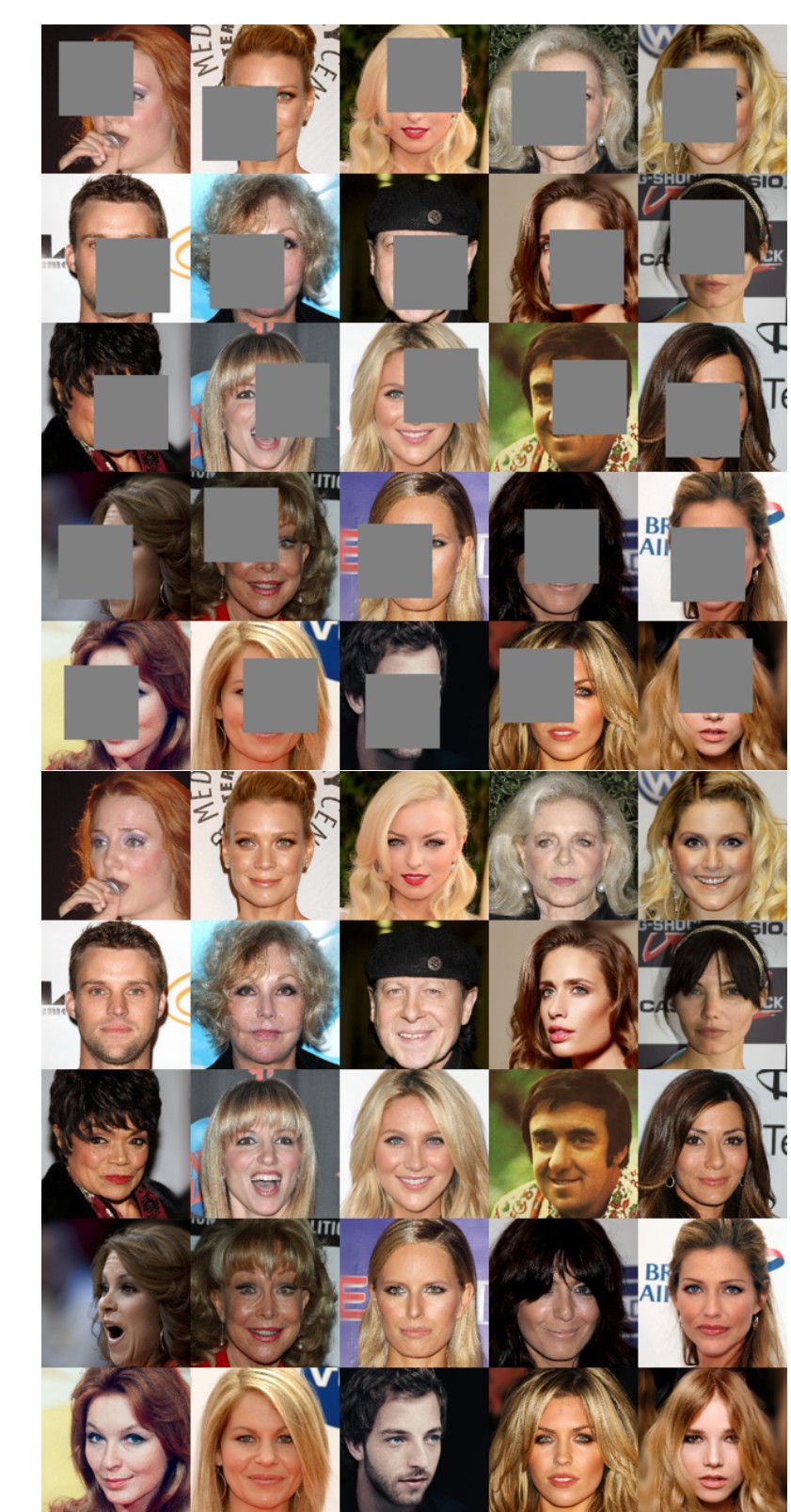

Figure 14: Figure illustrating **Non cherry picked** results for face inpainting

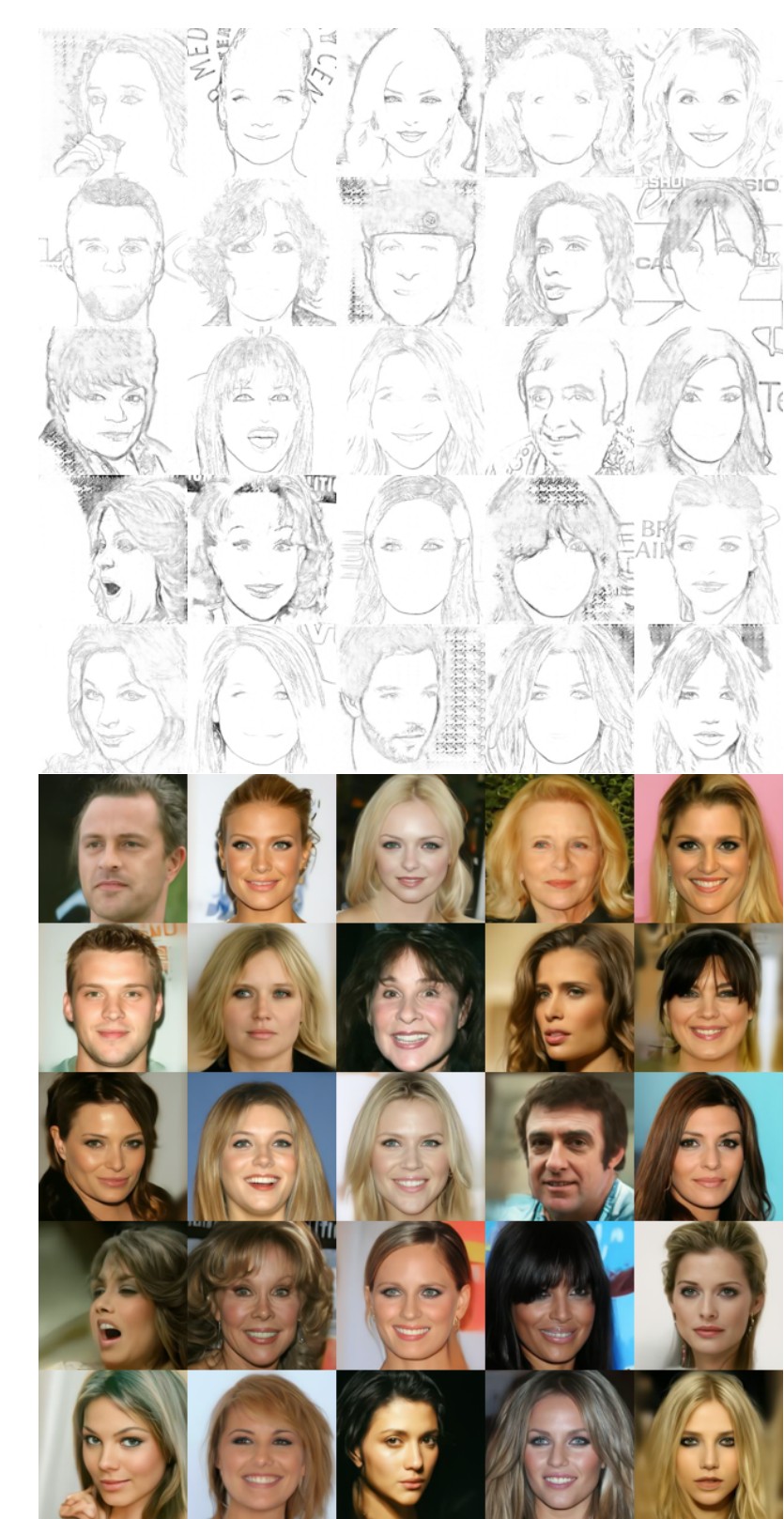

Figure 15: Figure illustrating **Non cherry picked** results for sketch to face synthesis

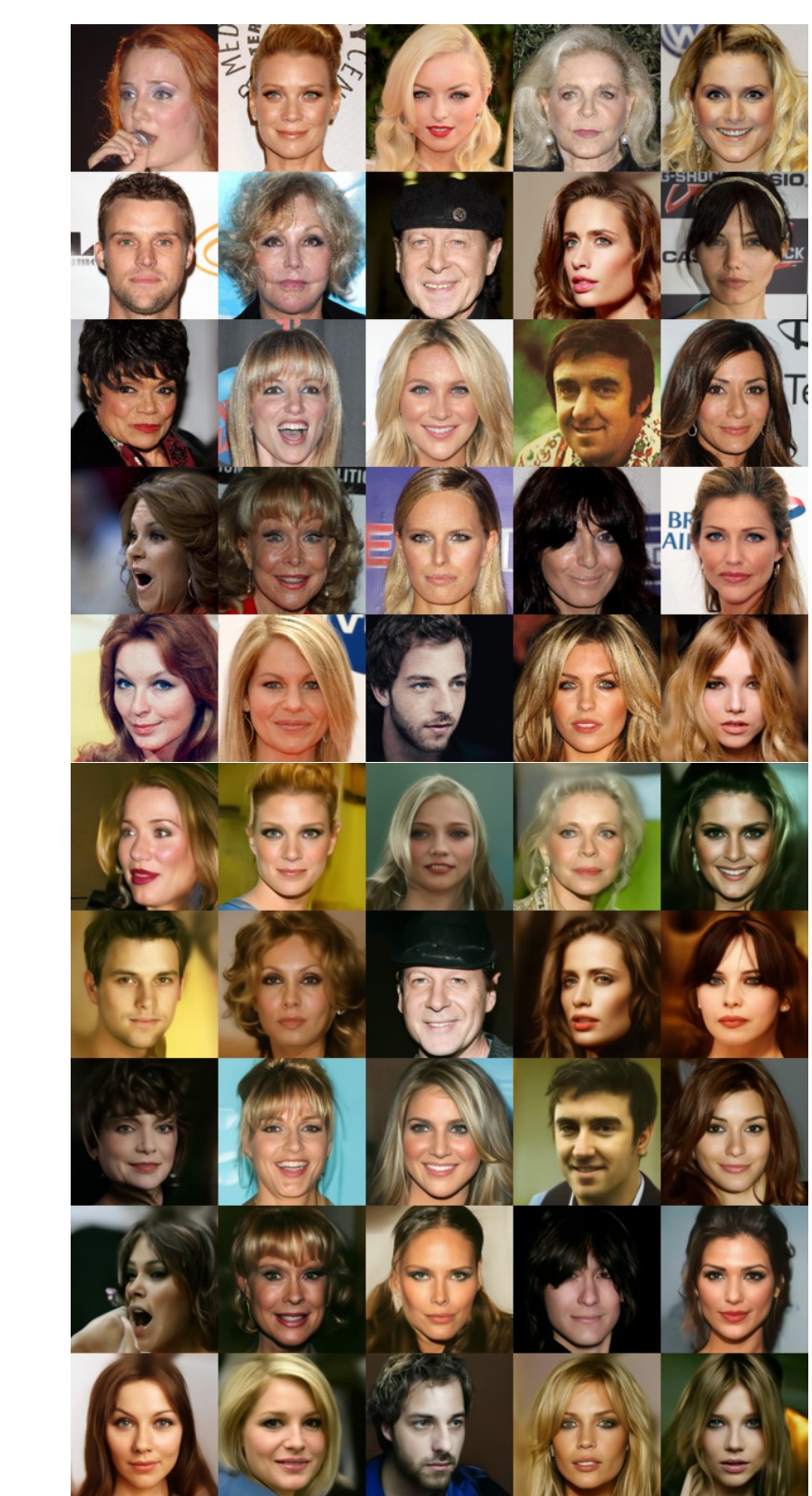

Figure 16: Figure illustrating **Non cherry picked** results for Face ID guidance

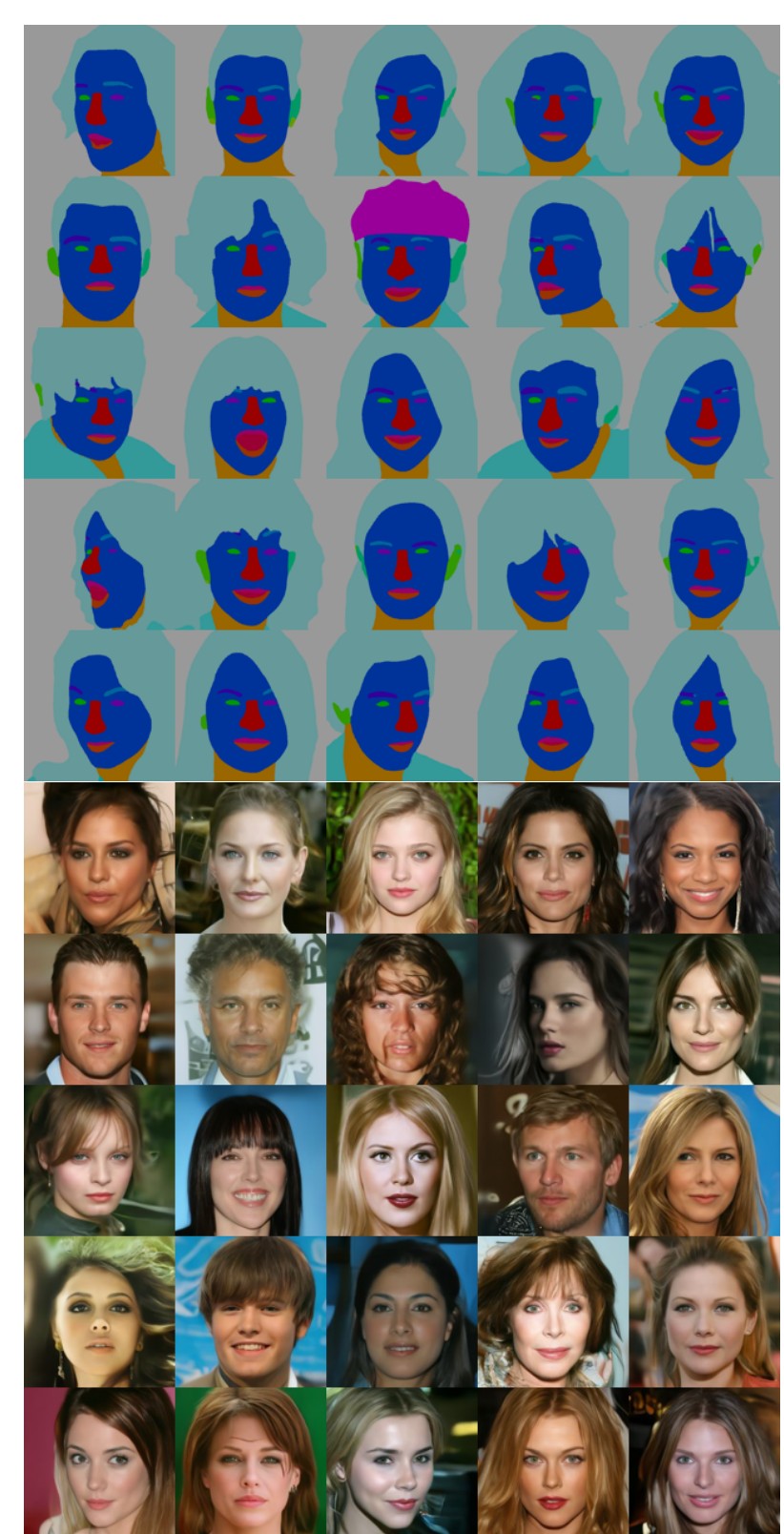

Figure 17: Figure illustrating **Non cherry picked** results for Face Parse Guidance

