# OpenReview forum: "DiffuseGuide: Guiding Diffusion Models Made Easy"
_ICLR.cc/2026/Conference — Submitted to ICLR 2026_

### Official Review · Reviewer_KV3q · 2025-10-25

**Soundness:** 1
**Presentation:** 1
**Contribution:** 2
**Rating:** 2
**Confidence:** 3

**Summary:**

This paper proposes **DiffuseGuide**, a training-free conditional guidance framework for diffusion models that avoids expensive backpropagation through the denoising network during sampling. The key observation is that guidance gradients can be approximated with respect to the **MMSE estimate** \( \hat{x}\_t \) and the **predicted noise** \( \epsilon_\theta(x\_t) \), enabling a **zeroth-order** optimization approach. This removes the major computational bottleneck of previous loss-guided sampling methods.

The method further introduces:

1. **Double-descent guidance**
   - Early timesteps: guide via noise gradients
   - Late timesteps: guide via MMSE gradients
2. **Automatic, gradient-dependent scale estimation**
   - Uses DOG-style dynamic learning-rate scheduling
   - Eliminates task-specific parameter tuning
3. **DiffuseAugment**
   - Differentiable augmentations applied to both prediction and condition
   - Prevents artifacts and improves perceptual fidelity

**Strengths:**

- Backpropagation-free sampling, enabling significantly lower inference compute while maintaining high guidance quality
- Works robustly across both linear and non-linear conditional tasks, outperforming prior training-free guidance methods

**Weaknesses:**

* **Unclear writing and missing implementation details**: The description of DiffuseAugment is insufficiently specified. It remains unclear *when* and *how* augmentations are applied, what exact operators are used per task, and how hyperparameters are chosen. The authors should provide task-specific augmentation configurations and a reproducible algorithm in the supplementary materials to ensure clarity and fairness in comparison.

* **Incorrect characterization of prior work (DPS and MGD)**: Table 1 incorrectly states that DPS and MGD cannot handle non-linear tasks, claiming a limitation to linear inverse problems. However, DPS only requires differentiability of the degradation operator (D), and MGD only requires differentiability of the guidance loss (L); both methods do not assume linearity. Indeed, DPS has demonstrated results on non-linear tasks such as phase retrieval and non-uniform deblurring, and MGD includes a nonlinear constraint example using a style loss (Frobenius norm between Gram matrices). These omissions should be corrected for an accurate representation of related work.

* **Unclear distinction from MGD**: The primary claimed contribution appears to be a backpropagation-free "double-descent classifier guidance". However, the second term in Eq. (11) can be rewritten via the chain rule as follows:
\(
\frac{\partial r}{\partial \epsilon_\theta(x_t)} =
\frac{\partial r}{\partial \hat{x}_t} \cdot
\frac{\partial \hat{x}_t}{\partial \epsilon_\theta(x_t)}
\)
where \(
\frac{\partial \hat{x}_t}{\partial \epsilon_\theta(x_t)}
\) is a constant derived from Eq. (10). In other words, this formulation effectively reduces to gradient descent on the clean MMSE estimate \(\hat{x}_t\) with a different scalar scaling, which is conceptually similar to MGD and also backpropagation-free. If I have misunderstood the distinction, I would appreciate it if the authors could correct my interpretation and clarify the fundamental differences from MGD.

**Questions:**

1. **Clarification on DiffuseAugment**
   Could the authors provide a more detailed description of DiffuseAugment, including the exact augmentation operators, hyperparameter settings, and task-specific configurations (ideally in an algorithmic form in the supplementary material)? *(See Weakness 1)*

2. **Correctness of claims about DPS and MGD**
   Since DPS and MGD are not limited to linear inverse problems and have demonstrated results on nonlinear tasks, could the authors revise Table 1 and the related discussion to accurately reflect the capabilities of these methods? *(See Weakness 2)*

3. **Distinction from MGD**
   Given that the second guidance term in Eq. (11) reduces mathematically to a scaled version of (\frac{\partial r}{\partial \hat{x}_t}) via the chain rule, could the authors clarify what fundamentally differentiates the proposed approach from MGD? If I have misunderstood the distinction, I would appreciate it if the authors could correct my interpretation and clarify the fundamental differences from MGD. *(See Weakness 3)*

4. **Effect of Different Guidance Components**
   The ablation in “Effect of Different Guidance Components” appears to be limited to a single task. Based on the results, (x_0)-only guidance performs best in linear tasks, while (\epsilon)-based guidance seems beneficial only in nonlinear scenarios.
   - Could the authors explain the intuition about this behavior?
   - Could the ablation be extended to additional tasks to confirm generality?

5. **Fairness and completeness of comparisons (Page 7)**
   The paper states that some methods are excluded because they "tackle only linear inverse problems", while loss-guided models are generic. However, ILVR is included only for super-resolution, despite being limited in capability.
   - If time allows, could the authors include those methods for the linear tasks on at least some overlapping tasks for fairness?
   - Even if their applicability is limited, showing head-to-head comparisons on shared tasks would help ensure an unbiased and complete evaluation.
   Currently, the selective inclusion of baselines risks appearing as if methods favorable to the proposed approach are emphasized while less favorable baselines are omitted.

---

### Official Review · Reviewer_adBf · 2025-10-28

**Soundness:** 2
**Presentation:** 2
**Contribution:** 2
**Rating:** 2
**Confidence:** 4

**Summary:**

The paper proposes a new inference-time guidance scheme for diffusion models, for linear and non-linear inverse problem solving. The method consists of three main components: 1) guidance without backpropagation: Considering the formula connecting the epsilon-prediction in DDPM to the denoiser output $\hat x_0 = x_t/\sqrt{\bar\alpha_t} + ( \sqrt{1-\bar\alpha_t} \epsilon_\theta(x_t) )/\sqrt{\bar\alpha_t}$, they add scaled versions of $\nabla_{\hat x_0} L(\hat x_0)$ and $\nabla_{\epsilon_\theta} L(\hat x_0)$ to the DDPM or DDPM updates for $x_{t-1}$ at each step of generation, with the scaling tuned such that the epsilon gradient is used more at high noise levels and the $x_0$ gradient is used more at low noise levels. 2) A new step size schedule for these gradient updates is implemented. 3) A new augmentation method for making the gradients more robust.

The resulting method achieves good performance on linear and non-linear inverse problems against baselines, does not require backpropagation during sampling, and little tuning across different problems.

**Strengths:**

- The method seems to work well and robustly solves many inverse problem tasks without task-specific tuning, and the ablations show that the $\hat x_0$-$\epsilon$ gradient idea and the augmentation trick both improve the results.
- The method is quite simple and probably quite easy to implement.
- There are lots of qualitative examples that the method works robustly
- The paper is mostly written in an easy-to-follow way

**Weaknesses:**

- I do not understand the motivation behind using the $\epsilon_\theta$ gradient: It appears that this is equivalent to using the $\hat x_0$ gradient, with a scaling based on the noise schedule. But if we are already scaling the $x_0$ gradient with the noise schedule, why not include it in it? It seems to me that $\frac{\partial r(\hat x, y)}{\partial \epsilon_\theta} = \frac{\partial r(\hat x, y)}{\partial \hat x} \frac{\partial \hat x}{\partial \epsilon_\theta} = \frac{\partial r(\hat x, y)}{\partial \hat x} (\frac{-\sqrt{1-\bar\alpha_t}}{\sqrt{\bar\alpha_t}})$. This is further scaled with $d\frac{1-\alpha_t}{\sqrt{\alpha_t}\sqrt{1-\bar\alpha_t}}$, resulting in total scaling of $-d\frac{(1-\alpha_t)}{\sqrt{\alpha_t \bar\alpha_t}}$. When added to the $\hat x$ term, this results in a gradient $\frac{\partial r(\hat x, y)}{\partial \hat x} (-c\sqrt{\alpha_{t-1}} - \frac{(1-\alpha_t)}{\sqrt{\alpha_t \bar\alpha_t}})$, which is further scaled with $\sigma_t^2$. Do the authors think this is correct, or am I missing something?
- Continuing on that, it would be better if all choices were motivated with some explicit reasoning about why that particular choice may help with improved results. Right now, the initial scheduling of the gradients and the gradient-dependent scaling factor estimate are both entirely heuristic. Adding two heuristic schedules on top of each other makes it seem like there would potentially be a much simpler way to frame the entire idea. It is also not clear why does the augmentation strategy help, although having the positive empirical results is a clear plus for the paper, to be clear.
- Some paper citations appear to be not entirely correct:
1) Graikos et al. [1] is not a paper on GANs, it is a paper on diffusion models.
2) It seems questionable whether the research on "plug-and-play" models was introduced in Nguyen et al. [2], since there were many papers before doing similar optimization in the GAN latent space before it (see the sentence "generating realistic-looking images by sampling in the latent space of a generator network..." in [2], and the referenced papers]). Although, Nuguyen et al. may have popularized the term in this context.
3) "All the aforementioned loss-guided posterior sampling... requires backpropagation". Bansal et al [3] also propose the "backward universal guidance" which works similarly to the method in this paper, and does not require backpropagation. Although the final method in the paper does also include backpropagation, it seems that there is nothing stopping one from just using the "backward guidance".
4) For DPS, Table 1 crosses out "nonlinear tasks", but the DPS paper explicitly considers nonlinear inverse problems. DPS is also not considered as a baseline for the nonlinear tasks.
5) "Universal diffusion guidance Aggarwal et al. (2018) extended this guidance process to Stable Diffusion and improved performance using forward–backward guidance" -> This should refer to Bansal et al?
- It seems that some of the baselines chosen for the paper are a bit weak, or not implemented in an entirely fair manner. DPS is dependent on the total scaling of the gradient, and if one wants to use a different amount of steps than the $T=1000$ in the original paper, the gradient scale should be recalibrated on a small validation set first. Also, DPS is applicable to nonlinear inverse problems as well. For the linear inverse problems, $\Pi GDM$ should also be a simple baseline that does not require hyperparameter tuning for different time step counts. What about using the backward method from the Universal Guidance paper? What about [4]? It is another similar method. Although a slightly different class of models, methods based on optimizing the generated sample like [5] also solve nonlinear inverse problems without backpropagating through the denoiser.

References:

[1] Graikos et al., "Diffusion models as plug-and-play priors"

[2] Nguyen et al., "Plug & play generative networks: Conditional iterative generation of images in latent space"

[3] Bansal et al., "Universal Guidance for Diffusion Models"

[4] Zhu et al.,  "Denoising Diffusion Models for Plug-and-Play Image Restoration"

[5]Mardani et al.,  "A VARIATIONAL PERSPECTIVE ON SOLVING INVERSE PROBLEMS WITH DIFFUSION MODELS"

**Questions:**

- What is the wall-clock time of the method, compared to the baselines? I would assume that it is faster than, e.g., DPS.
- Should Eq. 3. have an approximate sign $\approx$ instead of an equals sign?
- Why is Eq. 5 valid, in particular? Is this equation used for something in the method?
- It seems that Eq. 7 is not quite correct: It looks like this would be a Taylor expansion around $\mu$, but it is missing the $log p(y|\mu))$ term. The idea is that this is constant w.r.t. $x_{t-1}$ and thus does not affect the form of the reverse transition.
- What does it mean that a transition step is "valid"?
- What is "reciprocal distance"?

For now, I will start out with a reject due to the concerns raised in the weaknesses.

---

### Official Review · Reviewer_f223 · 2025-10-30

**Soundness:** 2
**Presentation:** 1
**Contribution:** 2
**Rating:** 4
**Confidence:** 2

**Summary:**

This paper proposed backpropagation-free, test-time guidance method to solve linear inverse problems. The authors proposed lightweight guidance solution by approximating direction of guidance with double-descent classifier guidance. In addition, they applied gradient-dependent scaling-factor estimate for inference sampling. The proposed method achieved best performances across various baselines.

**Strengths:**

-DiffuseGuide is able to several capabilities compared with baselines in Table 1. Refer from that table, DiffuseGuide is multi purpose and computationally cheap simultaneously.

-Double gradient descent method looks interesting. The authors proposed concrete derivation for that in equation (12)-(14). By using them, there is no need the calculation from posterior constraints which needs backpropagations of diffusion model.

-Various Experiments validates the capability of DiffuseGuide including Non-linear tasks. automatic scaling, linear tasks.

**Weaknesses:**

- Critical errors : line unchanged at line 407, line spacing at 457, different image at fig. 4 at the colorization column. Will it be acceptable?

- The main suggestion of this work is removing backpropagation routine to inference stage in diffusion sampling. It has to accelerate sampling phase by some amount. However, in main text (or even in supplementary), I couldn't find acceleration reports in terms of wall clock time for inference, training time saving, etc. I took the benefit from backpropagation-free as the main novelty of this work, so the authors need to clarify clear advantages from such benefits.

- Baselines including score-SDE, ILVR, DPS, MGD, are a bit outdated.

**Questions:**

I would like to know actual function of r(\cdot,y). What function could be a candidate for function r? In background section, it is described as measure of distance between x_t and condition y. However, there is no explicit list for function r. Linked with this, if I assume r function as L2 norm, how can we calculate dr/dx_t and dr/d epsilon (x_t)? Isn't it a backpropagated Jacobian of the given diffusion model? Could you explain this?

---

### Official Review · Reviewer_ZrDd · 2025-11-01

**Soundness:** 4
**Presentation:** 3
**Contribution:** 3
**Rating:** 8
**Confidence:** 3

**Summary:**

This paper proposes DiffuseGuide, a training-free inference-time guidance method for diffusion models that avoids backpropagation through the U-Net. The approach applies to both linear and non-linear inverse problems by combining double-descent guidance on the MMSE estimate and predicted noise. Extensive experiments on diverse tasks, such as inpainting, colorization, super-resolution, and deblurring, show consistent improvements over prior training-free baselines (DPS, MGD, Freedom) with better image quality.

**Strengths:**

- **Clear motivation and well-defined problem.** The paper correctly identifies a bottleneck in training-free conditional diffusion and the computational overhead of backpropagating through large U-Nets. The argument that zeroth-order guidance can unify linear and non-linear inverse problems is well framed.
- **Simple yet novel formulation** The proposed double-descent update and gradient-based scaling are conceptually simple but effectively address both early and late-stage guidance without backpropagation.
- **Comprehensive experiments.** The method is tested on a wide range of linear and non-linear tasks, consistently outperforming DPS, MGD, and Freedom across CelebA and ImageNet.
- **Solid theoretical grounding.** The method is supported by a clear mathematical formulation of the diffusion process and guidance, providing a coherent theoretical basis for the proposed inference-time updates.

**Weaknesses:**

- **Ablation.** Figure 6 explores only LPIPS trends with varying timesteps and augmentations. It would strengthen the paper to report the corresponding FID/PSNR.
- **Generalization to latent diffusion.** The method is not directly applicable to latent diffusion models, which limits its practical relevance for modern pretrained latent diffusion models.
- **Minor error.** The caption text around line 407 overlaps with the main body text.

**Questions:**

- The paper states that the MMSE estimate and the predicted noise are orthogonal, but since the MMSE estimate is derived from the predicted noise, this assumption is unclear. In addition, it would help to clarify why a separate dual-update formulation is needed when the gradient with respect to the predicted noise could, in principle, be expressed through the MMSE estimate. (I might be misunderstanding this point; if so, a clarification would be greatly appreciated.)

- Since eliminating U-Net backpropagation is a major claimed advantage, could the authors provide more concrete comparisons of inference time or computational cost to quantify this benefit?

---

> ### Author Response · Authors · 2025-12-03
> **Rebuattal for weakness**
>
> **W1.** We thank the reviewer for this suggestion, We include the PSNR/FID values here, we will update the main paper with the same
>
> | Method | 20       | 50      | 75     | 100         |
> |--------   |------------------------|------------------------|------------------------|----------------------------|
> | εₜ     | 23.84 / 84.99          | 22.22 / 82.28          | 21.52 / 79.35          | 21.03 / 141.28 (6 imgs)    |
> | full   | 28.40 / 76.74          | 28.41 / 78.06          | 28.42 / 76.27          | 28.43 / 73.94              |
> | x₀     | 28.54 / 54.83          | 28.44 / 51.71          | 28.36 / 46.42          | 28.25 / 43.80              |
>
> **W2.** Generalization to latent diffusion.
>
> As the reviewer rightly pointed out, our approach is not directly applicable to latent diffusion for all tasks. For latent diffusion models, diffuseguide works only for tasks that has a auxillary driving network, as an example we point out style-guided generation in Figure 1, where we can produce images adhering a particular style using a latent diffusion model. The main bottleneck here is the SD-VAE which causes gradients to become noisy and diminish in strength, losing quality for fine-grained tasks. However, for tasks that work at a semantic level, like style, ID guided generation, our approach still works and we could perform few-step hyper parameter optimization free guidance.
>
> **W3.** Minor error, we have updated the manuscript with the error corrected.

---

> > ### Author Response · Authors · 2025-12-03
> >
> > **Q1. Orthogonality**
> >
> > The orthogonality claim arises from two observations. First, the dot product between ε_t and x_0 is approximately zero across all timesteps in our experiments. Second, x_0 represents the underlying signal, while ε_t denotes the noise component at each timestep. In diffusion models, ε(t) corresponds to the Gaussian noise injected throughout the process. We believe the model learns components not already present in the signal, which explains why the ε_t direction improves generation quality.
> >
> > Separating the guidance into x_0-based and ε_t-based components also improves interpretability. Guidance using x_0 is ineffective in the early diffusion steps because the estimates are dominated by noise. In contrast, guidance using ε_t is more stable and meaningful in these early stages. The opposite occurs in later steps: the x_0 estimates become clean, and gradients with respect to them dominate the update.
> >
> > Aside from these interpretability benefits, we agree with the reviewer that the guidance can indeed be written using a single unified expression, and we will clarify this in the revised manuscript.

---

### Meta-Review · Area_Chair_aC5A · 2025-12-27

**Summary:**

All reviewers agreed that the method is simple and simple to implement, and agreed that it seems to work well. However, there are two major concerns that were left unaddressed by the authors, and that must at least be commented on:

1. **Are the baselines fair?** Reviewers f223, adBf and commented that baselines where outdated or unfairly implemented and characterized. Please see the suggestions by adBf for the steps in this direction.
2. **What is the computational budget?** All reviewers highlighted that the paper does not mention the wall-clock cost vs quality tradeoff inherent to posterior adaptation. This is an important data point to assess a new method.

**Reviewer Concerns:**

The point that the method works for latent diffusion was an important concern that was addressed. The concerns I mentioned above are the most important outstanding points, but given that the authors chose not to answer to most reviewers, there are more concerns standing out.

**Reviewer Scores:**

They would not have changed their scores, since the authors did not reply to 3/4 reviewers.

---

### Decision · Program_Chairs · 2026-01-26

Reject